# Forward Chaining Neural Network for Rule Induction

## Abstract

Inductive Logic Programming (ILP) learns logical rules from data, forming an interpretable machine learning model. Early-stage symbolic ILP systems perform outstandingly on small-scale tasks but suffer from combinatorial explosion. Emerging neuro-symbolic ILP methods demonstrate a certain degree of scalability and are more robust to noisy data. However, existing neuro-symbolic ILP methods are limited to constrained language biases, hampering further scalability. In this work, we propose Forward Chaining Neural Network (FCNN), a stochastic neural network that can learn logical rules under any language bias. FCNN relaxes all syntactically correct rules into continuous spaces and searches for the semantically correct solutions via gradient-based optimization. Experiments on standard evaluation tasks and recently proposed large-scale tasks show that FCNN outperforms existing methods.

## 1 Introduction

Inductive Logic Programming (ILP) aims to learn interpretable logic models (Muggleton & Raedt, 1994; Muggleton et al., 2012; Cropper et al., 2020a; Cropper & Dumancic, 2022; Zhang et al., 2024), typically represented by formal logics, which can generalize uniformly by learning from a small amount of entities. For example, logical formulas in first-order logic generalize to any entity (i.e., constant, object) uniformly, intrinsically endowed with robust generalization ability. ILP has successful applications in bioinformatics (Inoue et al., 2013; Kaalia et al., 2016), robotics Sammut et al. (2015); Cropper & Muggleton (2015); Antanas et al. (2015), physics-informed learning (Dai et al., 2017), program analysis Albarghouthi et al. (2017); Sivaraman et al. (2019); Bartha & Cheney (2019), ecology (Bohan et al., 2017), games (Legras et al., 2018; Cropper et al., 2020b), sequence processing (Evans et al., 2021; 2022; Gao et al., 2025) etc.

ILP originates from symbolic methods (Law et al., 2020; Cropper & Morel, 2021b; Cropper, 2022; Cropper & Hocquette, 2023), which often suffer from the combinatorial explosion problem (Cropper & Morel, 2021a) and from handling noisy data (Hocquette et al., 2024). Neuro-symbolic ILP methods (Evans & Grefenstette, 2018; Gao et al., 2024b), which leverage common techniques from Deep Learning (Bengio et al., 2017), such as continuous parameterization, probabilistic modeling, and gradient-based optimization, show great potential in scaling up. Symbolic methods can only surpass neuro-symbolic methods in small-scale tasks and noise-free settings (Glanois et al., 2022). Consequently, our work focuses on neuro-symbolic ILP, aiming to further scale up.

As an emerging research topic, existing neuro-symbolic ILP methods concentrate on limited language biases (i.e., syntax), impeding larger applications. For instance, existing works focus on learning unary and binary predicates, incapable of capturing hyper-relations involving multiple entities, which are helpful in real-world applications (Guan et al., 2019; Galkin et al., 2020; Wang et al., 2021). In addition, most neuro-symbolic ILP methods limit the number of body atoms to less than 2, incapable of learning more interpretable and more inference-efficient rules [1]. Consequently, our work aims to learn Horn rules, a type of rule widely used in ILP, under any language bias.

---

[1] Although a Horn rule $p \leftarrow q_1 \wedge q_2 \wedge q_3$ can be equivalently represented by two rules $p \leftarrow q_1 \wedge q^*, q^* \leftarrow q_2 \wedge q_3$, ILP models have to invent the dummy predicate $q^*$, which may be meaningless. In addition, inferring via the latter rules also introduce one extra forward chaining step, which may decrease inference efficiency.

Our main contribution is FCNN, a stochastic neural network that relaxes all syntactically correct Horn rules into continuous spaces and is optimized via gradient-based optimizers to find semantically correct hypotheses. The central component is a set of universal meta-rules. Significantly different from existing neuro-symbolic ILP, the universal meta-rules are not restricted to any specific language bias and do not require specific templates or candidate rules. The universal meta-rule enables direct learning of logical variables and atoms, instead of manually assigning variables, which shows better scalability when the predicates' arity and number of body atoms are larger.

Our main technical innovations are four-fold. First, to sample body atoms from FCNN's stochastic structures, we propose a linear sampling of body atoms, which mathematically is a sampling of multiple independent and non-identical distributed Bernoulli variables under an inequality constraint. Although naive rejection sampling can realize the sampling, the time usage will be significantly large if the constraint probability is too small (i.e., very low acceptance rate). Second, to estimate the gradient of FCNN, we propose a gradient estimator according to FCNN's specific stochastic structures. We theoretically prove the estimator's unbiasedness and experimentally verify its effectiveness. Third, we propose an algorithm for extracting rules from FCNN, in particular, extracting body atoms, and theoretically prove its soundness. Fourth, we prove that FCNN is complete for modeling Horn rules.

We conduct experiments on standard small-scale ILP tasks and recently proposed large-scale ILP tasks. Results indicate that FCNN performs better than existing methods and large language models (LLMs), though the latter recently show excellent reasoning ability in many other domains.

## 2 RELATED WORK

Neuro-symbolic ILP is an emerging area, which relaxes discrete hypothesis spaces into continuous spaces and optimizes parameters via gradient-based optimizers. Yang et al. (2017) propose a differentiable learning of logical rules, but limited to chain-like rules. Evans & Grefenstette (2018) design a relatively more general rule-level relaxation $\partial$ILP but limited to binary predicates and at most two body atoms. Difflog (Si et al., 2019) leverages another rule-level relaxation and requires a rule-candidate set, also restricted to limited language bias and intractable for large-scale ILP tasks in terms of memory consumption (Chen et al., 2025). Similar rule-level relaxations have been explored in $\partial$ILP-ST (Shindo et al., 2021), $\alpha$ILP (Shindo et al., 2023), and NEUMANN (Shindo et al., 2024), but require high-quality hypothesis space to obtain good performance (Shindo et al., 2021).

LRI (Campero et al., 2018) proposes a predicate-level relaxation, though benefiting from fine-grained continuous modeling, but requires task-specific templates and is thus not scalable. Several logic programming works (Rocktäschel & Riedel, 2017; Maene & Raedt, 2023), though supporting inductive logic programming by predicate-level relaxations, also leverage task-specific templates. To discard the dependence on specific templates, Glanois et al. (2022) provide a theoretically complete set of proto-rules and constructs the HRI model. However, these proto-rules can only be instantiated as Horn rules with unary & binary predicates and with at most two body atoms, i.e., restricted to a strong language bias. Gao et al. (2024a;b)'s DFORL learns same-head Horn programs, i.e., all the rules share the same head atom, but also only consider unary & binary predicates.

In contrast, our work proposes to use universal meta-rules, which can be seen as general templates (but not specific templates) that can be instantiated to any Horn rules, bypassing hand-crafted or syntactically restricted templates. In addition, almost all ILP systems assume the closed-world assumption (Reiter, 1981). Our work also considers the open-world assumption, which is a more realistic setting.

## 3 PRELIMINARIES

### 3.1 INDUCTIVE LOGIC PROGRAMMING

Horn rule is a class of logical rules in first-order logic [2] [3], which is widely used in artificial intelligence (Lloyd, 2012). Take $head(X_1, X_2) \leftarrow body_1(X_2, X_3, X_1) \wedge body_2(X_1) \wedge body_3(X_3, X_3)$

---

[2]We assume basic knowledge of first-order logic.

[3]We consider function-free Horn rules, as previous neuro-symbolic works do.

as an example. The $head, body_1, body_2, body_3$ are *predicates* with arity $2, 3, 1, 2$, respectively, i.e., they can take $2, 3, 1, 2$ arguments, respectively. The $X_1, X_2, X_3$ are logical *variables*, which are possible values of the arguments. The $head(X_1, X_2)$ is called *head atom*, and the $body_1(X_2, X_3, X_1), body_2(X_1), body_3(X_3, X_3)$ are called *body atoms*. These atoms are called *unground atoms* since they only have variables and not constants. Its counterpart is *ground atom*, also known as Horn *fact*, which is constituted by a predicate and several *constants*.

ILP learns a hypothesis $\mathcal{H}$, a set of Horn rules, from given background knowledge $\mathcal{B}$, positive target examples $\mathcal{E}^+$, and negative target examples $\mathcal{E}^-$ satifying

$$\mathcal{H} \wedge \mathcal{B} \vDash e^+, \quad \forall e^+ \in \mathcal{E}^+$$
$$\mathcal{H} \wedge \mathcal{B} \nvDash e^-, \quad \forall e^- \in \mathcal{E}^- \tag{1}$$

where $\mathcal{B}, \mathcal{E}^+, \mathcal{E}^-$ are sets of facts. Intuitively, eq. 1 states that, from given background facts, we can *infer* all positive examples based on the rules and not any negative examples. The *inference* is accomplished by forward chaining, introduced in the next section.

Closed-world assumption (CWA) and open-world assumption (OWA) (Reiter, 1981) are two different settings in ILP. Most ILP systems assume CWA, where the background knowledge is complete, i.e., $\mathcal{B}$ contains and only contains all *true* facts with background predicates. Our work also considers OWA, where $\mathcal{B}$ may be incomplete.

## 3.2 FORWARD CHAINING

*Forward chaining* is a deduction tool to infer all true facts from the background knowledge. Following the definition in Evans & Grefenstette (2018), for any set of ground atoms $\mathcal{A}$ and set of Horn rules $\mathcal{R}$, the facts deduced via one-step forward chaining is the following set

$$\mathcal{F}_{\mathcal{R},1}(\mathcal{A}) := \mathcal{A} \cup \left\{ a \,\middle|\, a \leftarrow a_1, \dots, a_k \in g(\mathcal{R}), \bigwedge_{l=1}^{k} a_l \in \mathcal{A} \right\} \tag{2}$$

where $g(\mathcal{R})$ denotes all the *ground rules* obtaining by substituting constants for all the variables in a rule of $\mathcal{R}$. Then, we recursively define the facts deduced through $t \geq 1$ steps $\mathcal{F}_{\mathcal{R},t}(\mathcal{A})$

$$\mathcal{F}_{\mathcal{R},t}(\mathcal{A}) := \mathcal{F}_{\mathcal{R},1}\big(\mathcal{F}_{\mathcal{R},t-1}(\mathcal{A})\big), \quad \text{where } \mathcal{F}_{\mathcal{R},0}(\mathcal{A}) := \mathcal{A} \tag{3}$$

We say the *fix point* is reached if, for some $T$, no new fact can be derived, i.e., $\mathcal{F}_{\mathcal{R},T}(\mathcal{A}) = \mathcal{F}_{\mathcal{R},T+1}(\mathcal{A})$. We define operator $\text{ForwardChaining}(\mathcal{B}, \mathcal{R})$ that takes background knowledge and a specific set of Horn rules as input and gives $\mathcal{F}_{\mathcal{R},T}(\mathcal{A})$ as output.

## 4 FORWARD CHAINING NEURAL NETWORK

A forward chaining neural network (FCNN) is established based on a set $\mathcal{U}$ of universal meta-rules. In contrast to previous works, these meta-rules are agnostic to any specific language bias and are expert-free. Hence, a universal meta-rule is simply an implication $\text{HEAD} \leftarrow \text{BODY}$, where HEAD is to be unified (i.e., matching) with an unground atom and BODY is to be unified with zero, one, or more unground atoms. The predicate in each atom may have any number of arguments, ensuring the modeling flexibility again. Besides, each universal meta-rule involves at most $V$ logical variables, i.e., the unified atoms can take these variables as arguments[4]. After training, we can extract interpretable rules from the universal meta-rules by substituting the unified head atom for HEAD, substituting the conjunction of the unified body atoms for BODY, and substituting the unified variables for the arguments.

Overall, the primary goal of FCNN is to learn the unification with respect to the universal meta-rules, involving unifying the unground atoms with HEAD, BODY and unifying the arguments of the unified atoms with the logical variables of the universal meta-rules. Similar to existing neuro-symbolic ILP methods, we attempt to leverage powerful continuous optimization to conduct the unification learning. See Algorithm 1 for the entire learning procedure.

---

[4]Slightly abusing the notion, we call the arguments of an atom's predicate the arguments of the atom. Similarly, we call the arity of an atom's predicate the arity of the atom, and the function $\text{arity}(\cdot)$ can be applied to atoms.

---

**Algorithm 1** Learning procedure

---

1: **INPUT:** background knowledge $\mathcal{B}$, target examples $\mathcal{E}^+, \mathcal{E}^-$, symbolic hyperparameters ($|\mathcal{U}|$, $B$, $V$, $|\mathcal{P}^{\mathrm{aux}}|, \ldots$), neural hyperparameters
2: initialize model parameters $\boldsymbol{\theta}$, set of head-candidate atoms $\mathcal{A}^{\mathrm{h}}$ (per CWA or OWA mode), set of body-candidate atoms $\mathcal{A}^{\mathrm{b}}$
3: **repeat**
4:    **for all** universal meta-rules **do**
5:       compute the probability for head atom unification               // Sec. 4.2.2
6:       compute the probability for body atom unification             // Sec. 4.2.3
7:       compute the probability for variable unification               // Sec. 4.2.4
8:    **end for**
9:    compute the probability for joint unification                   // Sec. 4.2.5
10:    linearly sample atoms and variables            // Sec. 4.3.2 and Algo. 2
11:    compute gradient $\widetilde{\nabla}_{\boldsymbol{\theta}}$ and optimize $\boldsymbol{\theta}$        // Sec. 4.3.1 & Eq. 7
12: **until** balanced accuracy achieves 100% or maximum iterations reached
13: extract interpretable rules from the model           // Sec. 4.4 and Algo. 3
14: **OUTPUT:** interpretable rules

---

We first define notations and concepts (Sec. 4.1), and then describe the continuous relaxations (Sec. 4.2), continuous optimization (Sec. 4.3), extraction of interpretable rules (Sec. 4.4), and hyperparameter analysis (Sec. 4.5).

### 4.1 NOTATIONS AND CONCEPTS

We use lowercase letters to denote predicates, e.g., $p, even$, and uppercase letters to denote hyperparameters (e.g., $V$). For positive integer $N$, we use $[N]$ to denote the set $\{1, \ldots, N\}$. We use the calligraphic font to denote sets, e.g., $\mathcal{P}, \mathcal{A}$. RV is the abbreviation for *random variable*. $[\ldots, \ldots, \ldots]$ denotes concatenation of vectors, and $\cdot$ denotes inner products.

Let $\mathcal{P}^{\mathrm{bk}}$, $\mathcal{P}^{\mathrm{aux}}$, and $\mathcal{P}^{\mathrm{tgt}}$ denote, respectively, the sets of background, auxiliary, and target predicates. The number of auxiliary predicates $|\mathcal{P}^{\mathrm{aux}}|$, as well as the maximum arity of an auxiliary predicate, are hyperparameters. Let $\mathcal{P} := \mathcal{P}^{\mathrm{bk}} \cup \mathcal{P}^{\mathrm{aux}} \cup \mathcal{P}^{\mathrm{tgt}}$ to denote the union of all predicates. Let the function $\mathrm{arity}(p)$ denote the arity of predicate $p \in \mathcal{P}$, and let $A$ to denote the maximum arity $\max_{p \in \mathcal{P}}(\mathrm{arity}(p))$. Since FCNN allows a predicate to occur multiple times in the body of an instantiated rule, we use $M$, referred to as multiplicity, to denote the maximum number of times a predicate is allowed to appear in any instantiated rule.

To learn the unifications, we define the set of *head-candidate atoms* $\mathcal{A}^{\mathrm{h}}(\mathcal{P}) := \{p(\mathrm{arg}_1^p, \ldots, \mathrm{arg}_{\mathrm{arity}(p)}^p) \,|\, p \in \mathcal{P}\}$ for each universal meta-rule. (Since the unifications are established meta-rule by meta-rule, we omit the meta-rule index $r \in [|\mathcal{U}|]$ for all candidate sets, embeddings, random variables defined in Section 4 to simplify notations. We will explicitly mention the meta-rule index when needed.) Each $\mathrm{arg}$ refers to an argument of the corresponding predicate, and is to be unified with the universal meta-rule's logical variables. The candidate atoms in $\mathcal{A}^{\mathrm{h}}(\mathcal{P})$ are to be unified with the meta-rule's HEAD. Similarly, we define the set of *body-candidate atoms* $\mathcal{A}^{\mathrm{b}}(\mathcal{P}, M) := \{p(\mathrm{arg}_1^{p,m}, \ldots, \mathrm{arg}_{\mathrm{arity}(p)}^{p,m}) \,|\, p \in \mathcal{P}, m \in [M]\}$ for each universal meta-rule. The definition reflects that each occurrence of the same predicate may take different arguments, enabling high flexibility of modeling. The candidate atoms in $\mathcal{A}^{\mathrm{b}}(\mathcal{P})$ are to be unified with the meta-rule's BODY. The dependence on $\mathcal{P}, M$ will be omitted in the following notations since it is fixed for each ILP task.

### 4.2 CONTINUOUS RELAXATION

To learn the unifications via continuous optimization, the symbolic units are relaxed into continuous spaces. Hence, we represent the universal meta-rules, including HEAD & BODY & logical variables, and candidate atoms, including predicates & arguments, by continuous embeddings (Sec. 4.2.1). Furthermore, to leverage gradient-based optimizers, FCNN's gradient should be tractable. Considering that it is difficult to make the forward chaining (Eq. 2, 3) differentiable with respect to

atom unification and variable unification[5], we attempt to unify the symbolic units in a probabilistic way (Sec. 4.2.2 - 4.2.5), which introduces stochastic structures into the FCNN. Although stochastic structures are not directly differentiable, we can estimate unbiased gradients (Sec. 4.3.1) via sampling (Sec. 4.3.2).

### 4.2.1 CONTINUOUS REPRESENTATION

For each universal meta-rule, we associate (learnable) embeddings $\boldsymbol{\nu}^{\mathrm{h}} \in \mathbb{R}^{d_{\mathrm{a}}}$ and $\boldsymbol{\nu}^{\mathrm{b}} \in \mathbb{R}^{d_{\mathrm{a}}}$ with its HEAD and BODY, respectively. Embeddings $\boldsymbol{\nu}_1, \ldots, \boldsymbol{\nu}_V \in \mathbb{R}^{d_{\mathrm{v}}}$ are assigned to the meta-rule's variables, reflecting that variables are unrelated across rules.

For each head-candidate atom and body-candidate atom, we design predicate embeddings, argument embeddings, and atom embeddings as follows. For predicate $p \in \mathcal{P}$, we associate an embedding $\boldsymbol{\eta}_p \in \mathbb{R}^{d_{\mathrm{p}}}$, revealing that predicates are shared across rules. For each head-candidate atom $a \in \mathcal{A}^{\mathrm{h}}$, we associate an embedding $\boldsymbol{\eta}_{a,i}^{\mathrm{h}} \in \mathbb{R}^{d_{\mathrm{v}}}$ for its $i$-th argument, where $i \in [\mathrm{arity}(a)]$. Similarly, for each body-candidate atom $a \in \mathcal{A}^{\mathrm{b}}$, we give an embedding $\boldsymbol{\eta}_{a,i}^{\mathrm{b}} \in \mathbb{R}^{d_{\mathrm{v}}}$ for its $i$-th argument. The argument embedding design reveals that the same predicate may take different arguments, depending on occurring in head or occurring in body and depending on the occurrence in body.

The atom embeddings are constructed from the predicates and arguments embeddings above via concatenation and (learnable) linear transformations. That is, each head-candidate atom $a \in \mathcal{A}^{\mathrm{h}}$ is assigned the embedding $\boldsymbol{\eta}_a^{\mathrm{h}} := \boldsymbol{W}^{\mathrm{h}} \cdot [\boldsymbol{\eta}_p, \boldsymbol{\eta}_{a,1}^{\mathrm{h}}, \ldots, \boldsymbol{\eta}_{a,A}^{\mathrm{h}}] \in \mathbb{R}^{d_{\mathrm{a}}}$, where, for $i > \mathrm{arity}(p)$, the argument embedding $\boldsymbol{\eta}_{a,i}^{\mathrm{h}}$ is defined as the all-zero vector of length $d_{\mathrm{v}}$, and $\boldsymbol{W}^h$ is a $d_{\mathrm{a}} \times (d_{\mathrm{p}} + A \cdot d_{\mathrm{v}})$ matrix. Similarly, we define the embedding of a body-candidate atom $a \in \mathcal{A}^{\mathrm{b}}$ as $\boldsymbol{\eta}_a^{\mathrm{b}} := \boldsymbol{W}^{\mathrm{b}} \cdot [\boldsymbol{\eta}_p, \boldsymbol{\eta}_{a,1}^{\mathrm{b}}, \ldots, \boldsymbol{\eta}_{a,A}^{\mathrm{b}}] \in \mathbb{R}^{d_{\mathrm{a}}}$. We remark that the two linear mappings $\boldsymbol{W}^{\mathrm{h}}, \boldsymbol{W}^{\mathrm{b}}$ are also independent across meta-rules.

Finally, we use $\boldsymbol{\theta}$ to collectively denote all the above model parameters. These various embeddings grant FCNN complete ability to learn any Horn rule, further demonstrated in the following sections.

### 4.2.2 PROBABILISTIC HEAD ATOM UNIFICATION

Recall that all the following unifications are meta-rule independent, in line with the property that head atoms are unrelated across rules, which ensures complete flexibility of our universal meta-rules. Specifically, to unify the head-candidate atoms with the meta-rule's HEAD, we define an RV $\mathrm{a}^{\mathrm{h}}$ over the set $\mathcal{A}^{\mathrm{h}}$, with the categorical distribution $P_{\boldsymbol{\theta}}(\mathrm{a}^{\mathrm{h}} = a) = \mathrm{softmax}\left(\boldsymbol{\eta}_a^{\mathrm{h}} \cdot \boldsymbol{\nu}^{\mathrm{h}} : a \in \mathcal{A}^{\mathrm{h}}\right)_a$, for $a \in \mathcal{A}^{\mathrm{h}}$, i.e., probabilistically unifying the head atoms. [6] Such a unification reflects the property that the Horn rule only allows one head atom.

Different from previous works (Campero et al., 2018; Glanois et al., 2022), where only predicates are unified via learning embeddings, we learn to unify the entire atoms that are constituted by predicates and arguments, the same for the following body atom unifications.

In addition, since $\mathcal{A}^{\mathrm{h}}$ contains background predicates, new background facts can be inferred via forward chaining, which follows the OWA. We refer to this setting as the OWA mode of FCNN. In the CWA setting, since all correct background facts are completely given, we restrict $\mathcal{A}^{\mathrm{h}}$ to involve only auxiliary and target predicates $\mathcal{P}^{\mathrm{aux}} \cup \mathcal{P}^{\mathrm{tgt}}$. We refer to this setting as FCNN's CWA mode.

### 4.2.3 PROBABILISTIC BODY ATOM UNIFICATION

To unify the body-candidate atoms with the BODY of each universal meta-rule, let $\mathbf{a}^{\mathrm{b}} := (\mathrm{a}_a^{\mathrm{b}} : a \in \mathcal{A}^{\mathrm{b}})$ be a vector of independent Bernoulli RVs with

$$P_{\boldsymbol{\theta}}(\mathrm{a}_a^{\mathrm{b}} = 1) := \sigma\left(\boldsymbol{\eta}_a^{\mathrm{b}} \cdot \boldsymbol{\nu}^{\mathrm{b}}\right), \forall a \in \mathcal{A}^{\mathrm{b}}, \tag{4}$$

where $\sigma$ is the sigmoid function. The RVs indicate whether the corresponding body-candidate atom is unified to the BODY.

---

[5]Existing works only consider predicate unification, which can be differentiable using restricted templates.
[6]We slightly abuse $a$ to denote both an element of $\mathcal{A}^{\mathrm{h}}$ (or $\mathcal{A}^{\mathrm{b}}$) and its index. The same below.

To reduce the computational cost, we limit the number of atoms that can be unified to each BODY to be at most $B$, as in symbolic ILP systems (e.g., Cropper & Morel (2021a)). Let $\mathsf{s} := \sum_{a=1}^{|\mathcal{A}^{\mathrm{b}}|} \mathsf{a}_a^{\mathrm{b}}$. Then, by the independence assumption and the Bayes' rule, we have

$$P_{\boldsymbol{\theta}}\left(\mathbf{a}^{\mathrm{b}}|\mathsf{s} \leq B\right) = \frac{\prod_{a=1}^{|\mathcal{A}^{\mathrm{b}}|} P_{\boldsymbol{\theta}}(\mathsf{a}_a^{\mathrm{b}}) \cdot \mathbf{1}_{\{\mathsf{s} \leq B\}}}{P_{\boldsymbol{\theta}}(\mathsf{s} \leq B)}, \tag{5}$$

where $\mathbf{1}_{\{\cdot\}}$ is the indicator function evaluating to one if the enclosed condition is satisfied and zero otherwise. This unification reveals that body atoms are permutation-equivalent in a Horn rule (since conjunction is).

The differentiability, with respect to $\boldsymbol{\theta}$, of the left-hand side of Eq. 5 requires a differentiable computing algorithm for the denominator of the right-hand side, which is the cumulative distribution function (CDF) of a Poisson binomial RV. Ahmed et al. (2023) propose an algorithm to compute the probability mass function (PMF) of Poisson binomial RV in a differentiable way. The CDF can then be computed by summing over the PMFs.

### 4.2.4 PROBABILISTIC VARIABLE UNIFICATION

After unifying the atoms, we now unify the meta-rules' variables with the arguments of the unified atoms. Again, recall that the variable unifications are meta-rule independent. To this end, let $\mathbf{v}^{\mathrm{h}} := (\mathsf{v}_{a,i}^{\mathrm{h}})_{a \in \mathcal{A}^{\mathrm{h}}, i \in [\mathrm{arity}(a)]}$ be a vector of independent RVs where each RV $\mathsf{v}_{a,i}^{\mathrm{h}}$ takes its values from the set $\{1, \ldots, V\}$ according to the categorical distribution $P_{\boldsymbol{\theta}}(\mathsf{v}_{a,i}^{\mathrm{h}} = v) = \mathrm{softmax}\left(\boldsymbol{\eta}_{a,i}^{\mathrm{h}} \cdot \boldsymbol{\nu}_1, \ldots, \boldsymbol{\eta}_{a,i}^{\mathrm{h}} \cdot \boldsymbol{\nu}_V\right)_v$, for $v \in [V]$. The RVs $\mathsf{v}_{a,i}^{\mathrm{h}}$ determine probabilistic unification between the $v$-th variable and the $i$-th argument of the head-candidate atom $a$ with probability $P_{\boldsymbol{\theta}}(\mathsf{v}_{a,i}^{\mathrm{h}} = v)$.

Similarly, for body atoms, we repeat the same procedure with $\mathbf{v}^{\mathrm{b}} := (\mathsf{v}_{a,i}^{\mathrm{b}})_{a \in \mathcal{A}^{\mathrm{b}}, i \in [|\mathrm{arity}(a)|]}$, where each RV $\mathsf{v}_{a,i}^{\mathrm{b}}$ is distributed according to $P_{\boldsymbol{\theta}}(\mathsf{v}_{a,i}^{\mathrm{b}} = v) = \mathrm{softmax}\left(\boldsymbol{\eta}_{a,i}^{\mathrm{b}} \cdot \boldsymbol{\nu}_1, \ldots, \boldsymbol{\eta}_{a,i}^{\mathrm{b}} \cdot \boldsymbol{\nu}_V\right)_v$.

This probabilistic variable unification reflects the following properties of arguments and variables: (a) arguments in one rule share the same set of variables, whether in the head or in the body, which interact with each other in forward chaining; (b) the same predicate may have different arguments in one rule and across rule; (c) arguments are independently unified to variables, ensuring again complete flexibility of our universal meta-rules.

Different from previous works (Campero et al., 2018; Glanois et al., 2022), where arguments are specified manually or by expert-provided restricted proto-rules, we directly learn to unify the variables with the arguments, improving the automatic scalability to higher arity.

### 4.2.5 PROBABILISTIC JOINT UNIFICATION

After defining the unifications for each universal meta-rules, now we can define the joint unification. Specifically, the joint atom unification with respect to HEAD, BODY over all meta-rules is assumed independent and is done according to the distribution $P_{\boldsymbol{\theta}}(\mathbf{a}) := \prod_{r \in [|\mathcal{U}|]} P_{\boldsymbol{\theta}}(\mathsf{a}_r^{\mathrm{h}}) \cdot P_{\boldsymbol{\theta}}(\mathsf{a}_r^{\mathrm{b}} \mid \sum_{a=1}^{|\mathcal{A}^{\mathrm{b}}|} \mathsf{a}_{r,a}^{\mathrm{b}} \leq B)$. Assuming independence again, the joint variable unification is done according to $P_{\boldsymbol{\theta}}(\mathbf{v}) := \prod_{r \in [|\mathcal{U}|]} P_{\boldsymbol{\theta}}(\mathbf{v}_r^{\mathrm{h}}) \cdot P_{\boldsymbol{\theta}}(\mathbf{v}_r^{\mathrm{b}})$. These new RVs with $r$ index are the same as above by adding back the index $r$.

### 4.3 CONTINUOUS OPTIMIZATION

After sampling $\mathbf{a}$ and $\mathbf{v}$, each universal meta-rule is instantiated to a specific Horn rule. Leveraging these rules, we can predict the truth values of target facts via forward chaining $\mathrm{ForwardChaining}(\mathcal{B}, \mathcal{R}(\mathbf{a}, \mathbf{v}))$, where $\mathcal{R}(\mathbf{a}, \mathbf{v})$ denotes the rule set instantiated according to $\mathbf{a}, \mathbf{v}$. Since the truth values are usually imbalanced for the target predicate, we use balanced accuracy (Brodersen et al., 2010) between *true* and *false* to evaluate the correctness of the instantiated rules, computed by $(TP/P + TN/N)/2$, where $P, N$ are the number of *true*s and *false*s corresponding

---

**Algorithm 2** Linear sampling of body atoms

---

1: **input:** Bernoulli parameter $P_{\boldsymbol{\theta}}(\mathsf{a}_a^{\mathrm{b}} = 1)$ for $a \in \mathcal{A}^{\mathrm{b}}$, body-candidate set $\mathcal{A}^{\mathrm{b}}$, maximum number of body atoms $B$
2: **output:** $\mathbf{a}^{\mathrm{b}} = (\mathsf{a}_a^{\mathrm{b}})_{a \in \mathcal{A}^{\mathrm{b}}} \sim P_{\boldsymbol{\theta}} \left( \mathbf{a}^{\mathrm{b}} | \mathsf{s} \le B \right)$
3: BODY = [], $j = B$
4: **for** $a = |\mathcal{A}^{\mathrm{b}}|$ **to** 1 **do**
5:    $\mathsf{a}_a^{\mathrm{b}} \sim \mathrm{Bernoulli} \left( P_{\boldsymbol{\theta}}(\mathsf{a}_a^{\mathrm{b}} = 1) \cdot P_{\boldsymbol{\theta}} \left( \sum_{k=1}^{a-1} \mathsf{a}_k^{\mathrm{b}} \le j - 1 \right) \Big/ P_{\boldsymbol{\theta}} \left( \sum_{k=1}^{a} \mathsf{a}_k^{\mathrm{b}} \le j \right) \right)$
6:    **if** $\mathsf{a}_a^{\mathrm{b}} = 1$ **then**
7:       $j = j - 1$,      BODY.append($\mathsf{a}_a^{\mathrm{b}}$)
8:    **end if**
9: **end for**

---

to all target ground atoms [7], $TP, TN$ are the number of correctly predicted *true*s and *false*s. Given a specific ILP task, the balanced accuracy is thus a non-differentiable function of $\mathbf{a}, \mathbf{v}$, which we denote by $f(\mathbf{a}, \mathbf{v})$.

To search for correct hypotheses, we maximize the following objective

$$L(\boldsymbol{\theta}) := \mathbb{E}_{\mathbf{a} \sim P_{\boldsymbol{\theta}}(\mathbf{a})} \left[ \mathbb{E}_{\mathbf{v} \sim P_{\boldsymbol{\theta}}(\mathbf{v})} [f(\mathbf{a}, \mathbf{v})] \right] + \lambda H(\mathbf{a}, \mathbf{v}) \tag{6}$$

to learn the unifications, where $H(\mathbf{a}, \mathbf{v})$ is an entropy regularization commonly used to encourage exploration. After optimizing, the distributions are expected to collapse to or approximate deterministic distributions, which deterministically instantiate universal meta-rules to specific rules that can entail the targets. Maximizing such an objective via gradient-based methods, where learnable parameters are in the sampling distribution and not in the function $f$, is feasible with the REINFORCE gradient estimation (Williams, 1992). Considering our proposed multi-stage unification, we design a nested variant, including a nested gradient estimator (Sec. 4.3.1), nested sampling strategy (Sec. 4.3.2), and entropy regularization (Sec. 4.3.3).

### 4.3.1 GRADIENT ESTIMATOR

Leveraging the REINFORCE technique (Williams, 1992) two times sequentially, we can rewrite the gradient $\nabla \mathbb{E}_{P_{\boldsymbol{\theta}}(\mathbf{a})}[\mathbb{E}_{P_{\boldsymbol{\theta}}(\mathbf{v})}[f(\mathbf{a}, \mathbf{v})]]$ as $\mathbb{E}_{P_{\boldsymbol{\theta}}(\mathbf{a})}[\mathbb{E}_{P_{\boldsymbol{\theta}}(\mathbf{v})}[f(\mathbf{a}, \mathbf{v})\nabla \log P_{\boldsymbol{\theta}}(\mathbf{a}, \mathbf{v})]]$. Then, using nested Monte Carlo estimator (Rainforth et al., 2018), we derive the following unbiased nested estimator

$$\widetilde{\nabla}_{\boldsymbol{\theta}} = \frac{1}{N^a} \sum_{i=1}^{N^a} \frac{1}{N^v} \sum_{j=1}^{N^v} \left( f \left( \mathbf{a}^{(i)}, \mathbf{v}^{(i,j)}(\mathbf{a}^{(i)}) \right) - b_{ij} \right) \nabla \log P_{\boldsymbol{\theta}} \left( \mathbf{a}^{(i)}, \mathbf{v}^{(i,j)}(\mathbf{a}^{(i)}) \right), \tag{7}$$

where $\mathbf{a}^{(i)} \sim P_{\boldsymbol{\theta}}(\mathbf{a}), \mathbf{v}^{(i,j)}(\mathbf{a}^{(i)}) \sim P_{\boldsymbol{\theta}}(\mathbf{v})$. The dependence of $\mathbf{v}^{(i,j)}$ on $\mathbf{a}^{(i)}$ corresponds to the fact that, after sampling head and body atoms, only the variables belonging to these atoms contribute to forward chaining. Therefore, the irrelevant variables are removed from the log probability, i.e., $\mathbf{v}^{(i,j)}(\mathbf{a}^{(i)})$ is a subset of $\mathbf{v}^{(i,j)} \sim P_{\boldsymbol{\theta}}(\mathbf{v})$ where only the relevant variables are retained.

$b_{ij}$ is called *baseline* introduced for reducing estimation variance. A powerful baseline is leave-one-out average, resulting in REINFORCE leave-one-out (RLOO) estimator (Salimans & Knowles, 2014; Kool et al., 2019; Richter et al., 2020), which tends to increase the probability of the samples evaluated to better-than-average $f$ and decrease otherwise.

This estimator is still unbiased, even after removing the irrelevant variables. The unbiasedness is proved in Appendix A.1.

**Proposition 4.1.** *The gradient estimator $\widetilde{\nabla}_{\boldsymbol{\theta}}$ is unbiased.*

### 4.3.2 SAMPLING STRATEGY

Like other REINFORCE-based estimators, the estimator $\widetilde{\nabla}_{\boldsymbol{\theta}}$ requires sampling from the distribution. Specifically, we need to first sample atoms $\mathbf{a} = ((\mathsf{a}_r^{\mathrm{h}})_{r \in [|\mathcal{U}|]}, (\mathsf{a}_r^{\mathrm{b}})_{r \in [|\mathcal{U}|]})$ and then sample

---

[7]By definition of forward chaining (Eq. 2), we need to define an set of constants. All target ground atoms refer to the atoms constructed by combining the target predicate with all combinations of constants.

---

**Algorithm 3** Extracting body atoms

---

1: **input:** Bernoulli parameter $P_{\boldsymbol{\theta}}(\mathsf{a}_a^{\mathrm{b}} = 1)$ for $a \in \mathcal{A}^{\mathrm{b}}$, body-candidate set $\mathcal{A}^{\mathrm{b}}$, maximum number of body atoms $B$
2: **output:** $\mathbf{a}^{\mathrm{b}} = \arg\max_{\mathbf{a}^{\mathrm{b}}} P_{\boldsymbol{\theta}}\left(\mathbf{a}^{\mathrm{b}} | \mathsf{s} \leq B\right)$
3: rank $P_{\boldsymbol{\theta}}(\mathsf{a}_a^{\mathrm{b}} = 1)$ for $a \in \mathcal{A}^{\mathrm{b}}$ in decreasing order
4: select all the $a$ where $P_{\boldsymbol{\theta}}(\mathsf{a}_a^{\mathrm{b}} = 1) > 0.5$, resulting in $\mathcal{A}^+$
5: **while** $|\mathcal{A}^+| > B$ **do**
6:     drop the element in $\mathcal{A}^+$ with the smallest $P_{\boldsymbol{\theta}}(\mathsf{a}_a^{\mathrm{b}} = 1)$
7: **end while**

---

$\mathbf{v} = ((\mathbf{v}_r^{\mathrm{h}})_{r \in [|\mathcal{U}|]}, (\mathbf{v}_r^{\mathrm{b}})_{r \in [|\mathcal{U}|]})$. Considering the independence of these RVs, sampling $\mathsf{a}_r^{\mathrm{h}}, \mathbf{v}_r^{\mathrm{h}}, \mathbf{v}_r^{\mathrm{b}}$ is straightforward. The difficulty arises from sampling $\mathbf{a}_r^{\mathrm{b}}$ by Eq. 5, where different Bernoulli RVs are sampled conditional on an inequality constraint. Existing works can handle equality constraints (Ahmed et al., 2023). We propose a similar sequential sampling process, under the inequality constraint, also achieving linear time complexity. See Algorithm 2 and Appendix A.2 for proof.

**Proposition 4.2.** *Algorithm 2 samples body atoms from Eq. 5 in time $\mathcal{O}(n)$.*

A naive rejection sampling works for a moderate-to-high probability of the inequality constraint; however, the time usage will be unacceptable if the probability is too small (i.e., very small acceptance probability).

### 4.3.3 ENTROPY REGULARIZATION

The term $\lambda H(\mathbf{a}, \mathbf{v})$ in Eq. 6 is the entropy regularization term, which encourages exploration by sampling more diverse samples. Since the RVs are assumed independent, their joint entropy $H(\mathbf{a}, \mathbf{v})$ is simply their sum. Since the entropy of Eq. 5 is hard to exactly compute, we replace it with the entropy of Eq. 4, which is a Bernoulli distribution. $\lambda$ is a hyperparameter to control the exploration and exploitation. Following common techniques, we exponentially decay $\lambda$ over training epochs.

### 4.4 EXTRACTING INTERPRETABLE RULES

To extract interpretable rules from FCNN, each argument takes the variable with the maximum probability, and each HEAD takes the head-candidate atom with the maximum probability. For each BODY, we also need to select the body-candidate atoms with the maximum probability of Eq. 5, which is not trivial. We propose Algorithm 3 to extract body atoms, which is proved in Appendix A.3.

The following theorem demonstrates the sufficient capacity of FCNN, proved in Appendix A.4.

**Theorem 4.3** (FCNN's Completeness). *The FCNN defined by $\boldsymbol{\theta}$ can model any finite set of Horn rules.*

### 4.5 HYPERPARAMETERS ANALYSIS

The hyperparameters of FCNN are divided into two parts, which we call *symbolic hyperparameters* and *neural hyperparameters*. Symbolic hyperparameters include the number of universal meta-rules $|\mathcal{U}|$, maximum arity of auxiliary predicates, number of auxiliary predicates, maximum number of body atoms $B$, maximum number of variables $V$. These hyperparameters are analogous to several symbolic ILP methods (Cropper & Morel, 2021a). Neural hyperparameters include optimizer hyperparameters, entropy regularization coefficient $\lambda$, embedding dimension $d_{\mathrm{p}}, d_{\mathrm{v}}, d_{\mathrm{a}}$, sample size of the gradient estimator $N^a, N^v$. These are common hyperparameters in Deep Learning. Overall, the hyperparameters do not involve expert knowledge, such as hand-crafted rule templates, which is in line with the proposal from Chen et al. (2025).

Table 1: Results on small-scale ILP tasks

| Task | | $\partial$ILP | LRI | HRI | DFORL | FCNN |
|---|---|---|---|---|---|---|
| Arithmetic | Even-Odd | 100 | 100 | 40 | 100 | 100 |
| | Even-Succ2 | 48.5 | 100 | 40 | 100 | 100 |
| | Fizz | 10 | 10 | 0 | 0 | 100 |
| | Buzz | 35 | 70 | 40 | 0 | 100 |
| Lists | Length | 92.5 | 100 | 0 | 100 | 100 |
| Family Tree | Grandparent | 96.5 | 100 | 100 | 100 | 100 |
| | Uncle | 70 | - | - | 100 | 100 |
| Graph | Adjacent to Red | 50.5 | 100 | 100 | 100 | 100 |
| | Two Children | 95 | 0 | 100 | 100 | 100 |
| | Graph Coloring | 94.5 | 0 | 100 | 100 | 100 |

Table 2: Results on large-scale ILP tasks

| Model | circle_1 | circle_2 | cong_1 | para_1 | para_2 | para_3 |
|---|---|---|---|---|---|---|
| LLM | $80.0 \pm 25.8$ | $75.0 \pm 26.4$ | $76.9 \pm 16.9$ | $93.3 \pm 16.1$ | $\mathbf{93.3 \pm 16.1}$ | $86.2 \pm 20.8$ |
| FCNN | $\mathbf{80.0 \pm 0.0}$ | $\mathbf{80.1 \pm 2.7}$ | $\mathbf{81.3 \pm 2.8}$ | $\mathbf{96.5 \pm 2.9}$ | $83.0 \pm 4.2$ | $\mathbf{98.5 \pm 1.2}$ |

| perp_1 | eqangle_1 | eqangle_2 | eqratio_1 | eqratio_2 | eqratio_3 | Avg. |
|---|---|---|---|---|---|---|
| $80.0 \pm 13.9$ | $62.2 \pm 7.0$ | $54.7 \pm 4.0$ | $64.4 \pm 23.3$ | $\mathbf{72.5 \pm 17.2}$ | $\mathbf{79.6 \pm 18.0}$ | $76.5$ |
| $\mathbf{83.4 \pm 0.0}$ | $\mathbf{72.5 \pm 6.2}$ | $\mathbf{67.1 \pm 4.4}$ | $\mathbf{68.0 \pm 8.8}$ | $67.6 \pm 5.5$ | $66.1 \pm 3.8$ | $\mathbf{78.7}$ |

## 5 EXPERIMENTS

We evaluate FCNN and compare it against existing methods on both small-scale datasets and recently proposed large-scale datasets. We also conduct ablation studies to reveal the effect of CWA/OWA mode, removing irrelevant variables in Eq. 7, and two important hyperparameters $N^a, N^v$. See Appendix B for implementation details.

### 5.1 EXPERIMENTS ON SMALL-SCALE DATASETS

First, we experiment on standard ILP datasets proposed by Evans & Grefenstette (2018). The dataset contains 20 ILP tasks, involving arithmetic, list relation, family tree, and graphs. These tasks can be perfectly solved in the CWA setting with unary & binary predicates and with rules possessing less than 2 body atoms. Therefore, this small-scale datasets cannot fully evaluate FCNN's performance, but can compare FCNN with existing methods. We set CWA mode for FCNN in these tasks.

Table 1 show the results on the tasks not fully solved by existing works. Following previous works, the reported metric is the success rate over at least 10 random seeds. A trial is considered successful if, on validation set, the mean squared error between the truth value (of the target facts) and the predicted truth value is less than 1e-4. See Appendix C.1 for full results. FCNN can solve all of the 20 tasks, in particular, the hardest Fizz and Buzz tasks that require learning complicated recursive rules.

### 5.2 EXPERIMENTS ON LARGE-SCALE DATASETS

Second, we test on a recently proposed large-scale dataset GeoILP, which is infeasible for existing methods due to their restricted language bias or insufficient handling of high-arity predicates (Chen et al., 2025). We use the large language model (LLM) to compare against FCNN. The former shows excellent reasoning ability in many other reasoning domains. Specifically, we use DeepSeek-V3.1-Terminus [8] (thinking mode) (DeepSeek-AI, 2024), one of the latest and best reasoning LLMs. Details of using LLM to solve ILP tasks are given in Appendix B.2.

---

[8] https://huggingface.co/deepseek-ai/DeepSeek-V3.1-Terminus

Table 3: Comparison of CWA and OWA mode

| Task | CWA | OWA |
|------|-----|-----|
| circle_1 | **80.0 ± 0.0** | 79.3 ± 2.2 |
| cong_1 | 81.3 ± 2.8 | **81.8 ± 0.6** |
| para_1 | **96.5 ± 2.9** | 93.8 ± 6.4 |
| perp_1 | 83.4 ± 0.0 | **84.1 ± 2.2** |
| Avg. | **85.3** | 84.8 |

Table 4: Effect of removing irrelevant vars

| Task | remove | retain |
|------|--------|--------|
| circle_1 | **80.0 ± 0.0** | **80.0 ± 0.0** |
| cong_1 | 81.3 ± 2.8 | **81.6 ± 2.7** |
| para_1 | **96.5 ± 2.9** | 92.7 ± 7.0 |
| perp_1 | **83.4 ± 0.0** | **83.4 ± 0.0** |
| Avg. | **85.3** | 84.4 |

We focus on the basic level of GeoILP, involving predicates of up to 8 arguments and the OWA setting. Due to the high arity, the hypothesis space is far larger than the small-scale tasks, due to the combinatorial explosion in terms of arity (Cropper & Morel, 2021a). Table 2 show the results on 12 basic-level tasks. We report balanced accuracy (see Sec. 4.3) on validation set. All results are averaged over 10 random seeds. FCNN outperforms LLM on 9 out of 12 tasks, surpassing LLM by 2.2 on average. In addition, we found LLM suffers from extremely high variance on all tasks, revealing that LLM is not a robust ILP solver.

### 5.3 ABLATION STUDY AND HYPERPARAMETER ANALYSIS

We also conduct ablation studies to reveal the effect of CWA/OWA mode, of removing irrelevant variables in Eq. 7, and of the two important hyperparameters $N^a, N^v$. When studying one of the effects, we keep all other hyperparameters the same.

Table 3 shows the comparison between the CWA mode and the OWA mode. Results show that the two modes don't have a significant performance gap. The CWA mode surpasses the OWA mode by 0.5 on average. We argue that the larger hypothesis space in OWA mode increases the optimization difficulty. See Appendix C.2 for the CWA/OWA comparison on more tasks.

Table 4 shows the effect of removing irrelevant variables in Eq. 7. It is experimentally proven that FCNN can benefit from removing irrelevant variables, which improves balanced accuracy by 0.9 on average. In addition, removing irrelevant variables is nearly never worse than retaining them.

Figure 1 (in Appendix C) illustrates how balanced accuracy varies with variable sample size $N^v$ and atom sample size $N^a$. For a fair comparison, we keep the total number of training samples the same (Figure 1.a & Figure 1.c), i.e., increasing the sample size will decrease the number of training steps accordingly. We found that a large sample size can cause high variance and typically decrease the average balanced accuracy. To explore the reason, we keep the total number of training steps the same (Figure 1.b & Figure 1.d). In this case, a large sample size can improve the balanced accuracy and the variances typically do not increase obviously, even decrease on some tasks. The two phenomena above justifies that the superior performance of large sample sizes requires sufficient training steps.

See Appendix C.3 for comprehensive sensitivity analysis on other hyperparameters. See Appendix C.4 for time usage comparison.

## 6 CONCLUSION

Inductive Logic Programming (ILP) learns interpretable logical rules but struggles with combinatorial explosion. Neuro-symbolic ILP alleviates these issues, yet is restricted by narrow language biases. We propose the Forward Chaining Neural Network (FCNN), a new stochastic neural network for neuro-symbolic ILP that can learn any set of Horn rules under any language bias. Experiments show FCNN can achieve the best performance on both small-scale and large-scale ILP datasets.

## REPRODUCIBILITY STATEMENT

We have uploaded the code for experiments to the review system. All the implementation details, including hyperparameters, training procedure, and forward chaining, can be found in the code. Our code follows the reproducibility guide given by PyTorch. The code will also be publicly available.

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

## A PROOFS

### A.1 UNBIASEDNESS OF THE PROPOSED GRADIENT ESTIMATOR

Now we prove the unbiasedness of our proposed gradient estimator (Eq. 7).

$$\mathbb{E}_{P_{\boldsymbol{\theta}}(\mathbf{a})}\left[\mathbb{E}_{P_{\boldsymbol{\theta}}(\mathbf{v})}\left[\widetilde{\nabla}_{\boldsymbol{\theta}}\right]\right]$$

$$= \mathbb{E}_{P_{\boldsymbol{\theta}}(\mathbf{a})}\left[\mathbb{E}_{P_{\boldsymbol{\theta}}(\mathbf{v})}\left[\frac{1}{N^a}\sum_{i=1}^{N^a}\frac{1}{N^v}\sum_{j=1}^{N^v} f\left(\mathbf{a}^{(i)}, \mathbf{v}^{(i,j)}(\mathbf{a}^{(i)})\right)\nabla\log P_{\boldsymbol{\theta}}\left(\mathbf{a}^{(i)}, \mathbf{v}^{(i,j)}(\mathbf{a}^{(i)})\right)\right]\right]$$

$$= \frac{1}{N^a}\sum_{i=1}^{N^a}\frac{1}{N^v}\sum_{j=1}^{N^v}\mathbb{E}_{P_{\boldsymbol{\theta}}(\mathbf{a})}\left[\mathbb{E}_{P_{\boldsymbol{\theta}}(\mathbf{v})}\left[f\left(\mathbf{a}^{(i)}, \mathbf{v}^{(i,j)}(\mathbf{a}^{(i)})\right)\nabla\log P_{\boldsymbol{\theta}}\left(\mathbf{a}^{(i)}, \mathbf{v}^{(i,j)}(\mathbf{a}^{(i)})\right)\right]\right]$$

$$= \frac{1}{N^a}\sum_{i=1}^{N^a}\frac{1}{N^v}\sum_{j=1}^{N^v}\mathbb{E}_{P_{\boldsymbol{\theta}}(\mathbf{a})}\left[\mathbb{E}_{P_{\boldsymbol{\theta}}(\mathbf{v})}\left[f\left(\mathbf{a}^{(i)}, \mathbf{v}^{(i,j)}(\mathbf{a}^{(i)})\right)\nabla\log P_{\boldsymbol{\theta}}\left(\mathbf{a}^{(i)}, \mathbf{v}^{(i,j)}\right)\right]\right]$$

$$- \frac{1}{N^a}\sum_{i=1}^{N^a}\frac{1}{N^v}\sum_{j=1}^{N^v}\mathbb{E}_{P_{\boldsymbol{\theta}}(\mathbf{a})}\left[\mathbb{E}_{P_{\boldsymbol{\theta}}(\mathbf{v})}\left[f\left(\mathbf{a}^{(i)}, \mathbf{v}^{(i,j)}(\mathbf{a}^{(i)})\right)\nabla\log P_{\boldsymbol{\theta}}\left(\mathbf{a}^{(i)}, \mathbf{v}_{-}^{(i,j)}\right)\right]\right]$$

We ignore the baseline $b_{ij}$ for simplicity since its unbiasedness has already been proved in RLOO estimator (Salimans & Knowles, 2014; Kool et al., 2019; Richter et al., 2020).

In step 3, $\mathbf{v}_{-}^{(i,j)}$ denotes the variables in $\mathbf{v}^{(i,j)}$ but not in $\mathbf{v}^{(i,j)}(\mathbf{a}^{(i)})$. Step 3 is justified by the independence of RVs. Since $\mathbf{v}^{(i,j)}(\mathbf{a}^{(i)})$ and $\mathbf{v}_{-}^{(i,j)}$ are independent, the expectation in the second term can be decomposed into

$$\mathbb{E}_{\mathbf{v}_{-}^{(i,j)}}\left[\nabla\log P_{\boldsymbol{\theta}}\left(\mathbf{a}^{(i)}, \mathbf{v}_{-}^{(i,j)}\right)\right]\cdot\mathbb{E}_{\mathbf{v}^{(i,j)}(\mathbf{a}^{(i)})}\left[f\left(\mathbf{a}^{(i)}, \mathbf{v}^{(i,j)}(\mathbf{a}^{(i)})\right)\right]$$

The first term is zero by the conventional REINFORCE technique.

Therefore,

$$\mathbb{E}_{P_{\boldsymbol{\theta}}(\mathbf{a})}\left[\mathbb{E}_{P_{\boldsymbol{\theta}}(\mathbf{v})}\left[\widetilde{\nabla}_{\boldsymbol{\theta}}\right]\right]$$

$$= \frac{1}{N^a}\sum_{i=1}^{N^a}\frac{1}{N^v}\sum_{j=1}^{N^v}\mathbb{E}_{P_{\boldsymbol{\theta}}(\mathbf{a})}\left[\mathbb{E}_{P_{\boldsymbol{\theta}}(\mathbf{v})}\left[f\left(\mathbf{a}^{(i)}, \mathbf{v}^{(i,j)}(\mathbf{a}^{(i)})\right)\nabla\log P_{\boldsymbol{\theta}}\left(\mathbf{a}^{(i)}, \mathbf{v}^{(i,j)}\right)\right]\right]$$

$$= \frac{1}{N^a}\sum_{i=1}^{N^a}\frac{1}{N^v}\sum_{j=1}^{N^v}\mathbb{E}_{P_{\boldsymbol{\theta}}(\mathbf{a})}\left[\mathbb{E}_{P_{\boldsymbol{\theta}}(\mathbf{v})}\left[f\left(\mathbf{a}^{(i)}, \mathbf{v}^{(i,j)}\right)\nabla\log P_{\boldsymbol{\theta}}\left(\mathbf{a}^{(i)}, \mathbf{v}^{(i,j)}\right)\right]\right]$$

$$= \mathbb{E}_{P_{\boldsymbol{\theta}}(\mathbf{a})}\left[\mathbb{E}_{P_{\boldsymbol{\theta}}(\mathbf{v})}\left[\frac{1}{N^a}\sum_{i=1}^{N^a}\frac{1}{N^v}\sum_{j=1}^{N^v} f\left(\mathbf{a}^{(i)}, \mathbf{v}^{(i,j)}\right)\nabla\log P_{\boldsymbol{\theta}}\left(\mathbf{a}^{(i)}, \mathbf{v}^{(i,j)}\right)\right]\right]$$

The step before the last step removes the dependence of $\mathbf{v}^{(i,j)}$ on $\mathbf{a}^{(i)}$ since the irrelevant variables do not affect the forward chaining, as they do not appear in the instantiated rules. Therefore, the value of $f$ won't change. The last step corresponds to the canonical REINFORCE estimator with nested Monte Carlo estimation (Rainforth et al., 2018), which completes the proof.

### A.2 CORRECTNESS OF ALGORITHM 2

In this section, we prove that the samples obtained from Algorithm 2 follow the left-hand side of Eq. 5. Mathematically, this problem is similar to the Proposition 2 in (Ahmed et al., 2023), except

that the constraint is an inequality in our case. Our proof below is analogous to the proof in (Ahmed et al., 2023).

The left-hand side of Eq. 5 can be decomposed as

$$\sum_{a=1}^{|\mathcal{A}^{\mathrm{b}}|} P\left(\mathsf{a}_a^{\mathrm{b}} \;\middle|\; \mathsf{a}_{a+1}^{\mathrm{b}}, \ldots, \mathsf{a}_{|\mathcal{A}^{\mathrm{b}}|}^{\mathrm{b}}, \sum_{k=1}^{|\mathcal{A}^{\mathrm{b}}|} \mathsf{a}_k^{\mathrm{b}} \le B\right).$$

Next, we will further decompose this conditional sampling probability.

Assume that RVs $\mathsf{a}_{a+1}^{\mathrm{b}}, \ldots, \mathsf{a}_{|\mathcal{A}^{\mathrm{b}}|}^{\mathrm{b}}$ have been sampled according to Algorithm 2 and that their values are $\hat{\mathsf{a}}_{a+1}^{\mathrm{b}}, \ldots, \hat{\mathsf{a}}_{|\mathcal{A}^{\mathrm{b}}|}^{\mathrm{b}}$. Let $j = B - \sum_{k=a+1}^{|\mathcal{A}^{\mathrm{b}}|} \hat{\mathsf{a}}_k^{\mathrm{b}}$. Consider the following conditional sampling probability

$$P\left(\mathsf{a}_a^{\mathrm{b}} = 1 \;\middle|\; \mathsf{a}_{a+1}^{\mathrm{b}} = \hat{\mathsf{a}}_{a+1}^{\mathrm{b}}, \ldots, \mathsf{a}_{|\mathcal{A}^{\mathrm{b}}|}^{\mathrm{b}} = \hat{\mathsf{a}}_{|\mathcal{A}^{\mathrm{b}}|}^{\mathrm{b}}, \sum_{k=1}^{|\mathcal{A}^{\mathrm{b}}|} \mathsf{a}_k^{\mathrm{b}} \le B\right)$$

$$= P\left(\mathsf{a}_a^{\mathrm{b}} = 1 \;\middle|\; \mathsf{a}_{a+1}^{\mathrm{b}} = \hat{\mathsf{a}}_{a+1}^{\mathrm{b}}, \ldots, \mathsf{a}_{|\mathcal{A}^{\mathrm{b}}|}^{\mathrm{b}} = \hat{\mathsf{a}}_{|\mathcal{A}^{\mathrm{b}}|}^{\mathrm{b}}, \sum_{k=1}^{a} \mathsf{a}_k^{\mathrm{b}} \le j\right)$$

$$= \frac{P\left(\mathsf{a}_a^{\mathrm{b}} = 1, \mathsf{a}_{a+1}^{\mathrm{b}} = \hat{\mathsf{a}}_{a+1}^{\mathrm{b}}, \ldots, \mathsf{a}_{|\mathcal{A}^{\mathrm{b}}|}^{\mathrm{b}} = \hat{\mathsf{a}}_{|\mathcal{A}^{\mathrm{b}}|}^{\mathrm{b}}, \sum_{k=1}^{a} \mathsf{a}_k^{\mathrm{b}} \le j\right)}{P\left(\mathsf{a}_{a+1}^{\mathrm{b}} = \hat{\mathsf{a}}_{a+1}^{\mathrm{b}}, \ldots, \mathsf{a}_{|\mathcal{A}^{\mathrm{b}}|}^{\mathrm{b}} = \hat{\mathsf{a}}_{|\mathcal{A}^{\mathrm{b}}|}^{\mathrm{b}}, \sum_{k=1}^{a} \mathsf{a}_k^{\mathrm{b}} \le j\right)}$$

$$= \frac{P\left(\mathsf{a}_a^{\mathrm{b}} = 1, \mathsf{a}_{a+1}^{\mathrm{b}} = \hat{\mathsf{a}}_{a+1}^{\mathrm{b}}, \ldots, \mathsf{a}_{|\mathcal{A}^{\mathrm{b}}|}^{\mathrm{b}} = \hat{\mathsf{a}}_{|\mathcal{A}^{\mathrm{b}}|}^{\mathrm{b}}, \sum_{k=1}^{a-1} \mathsf{a}_k^{\mathrm{b}} \le j - 1\right)}{P\left(\mathsf{a}_{a+1}^{\mathrm{b}} = \hat{\mathsf{a}}_{a+1}^{\mathrm{b}}, \ldots, \mathsf{a}_{|\mathcal{A}^{\mathrm{b}}|}^{\mathrm{b}} = \hat{\mathsf{a}}_{|\mathcal{A}^{\mathrm{b}}|}^{\mathrm{b}}, \sum_{k=1}^{a} \mathsf{a}_k^{\mathrm{b}} \le j\right)}$$

$$= \frac{P\left(\mathsf{a}_a^{\mathrm{b}} = 1\right) \cdot P\left(\mathsf{a}_{a+1}^{\mathrm{b}} = \hat{\mathsf{a}}_{a+1}^{\mathrm{b}}, \ldots, \mathsf{a}_{|\mathcal{A}^{\mathrm{b}}|}^{\mathrm{b}} = \hat{\mathsf{a}}_{|\mathcal{A}^{\mathrm{b}}|}^{\mathrm{b}}\right) \cdot P\left(\sum_{k=1}^{a-1} \mathsf{a}_k^{\mathrm{b}} \le j - 1\right)}{P\left(\mathsf{a}_{a+1}^{\mathrm{b}} = \hat{\mathsf{a}}_{a+1}^{\mathrm{b}}, \ldots, \mathsf{a}_{|\mathcal{A}^{\mathrm{b}}|}^{\mathrm{b}} = \hat{\mathsf{a}}_{|\mathcal{A}^{\mathrm{b}}|}^{\mathrm{b}}\right) \cdot P\left(\sum_{k=1}^{a} \mathsf{a}_k^{\mathrm{b}} \le j\right)}$$

$$= \frac{P(\mathsf{a}_a^{\mathrm{b}} = 1) \cdot P\left(\sum_{k=1}^{a-1} \mathsf{a}_k^{\mathrm{b}} \le j - 1\right)}{P\left(\sum_{k=1}^{a} \mathsf{a}_k^{\mathrm{b}} \le j\right)}$$

The second last step benefits from RVs' independence. The final step corresponds to the Bernoulli parameter in Algorithm 2.

### A.3 Correctness of Algorithm 3

Algorithm 3 corresponds to the following problem: given Bernoulli RVs $\mathsf{b}^{(i)}$ with different Bernoulli parameters $p_1, \ldots, p_n$, how to find a realization that maximizes their joint probability under the constraints $\sum_{i=1}^{n} \mathsf{b}^{(i)} \le B$. Their joint log probability is

$$\log \prod_{i=1}^{n} p_i^{\mathsf{b}^{(i)}} (1 - p_i)^{1 - \mathsf{b}^{(i)}} = \sum_{i=1}^{n} \mathsf{b}^{(i)} \log p_i + (1 - \mathsf{b}^{(i)}) \log(1 - p_i)$$

$$= \sum_{i=1}^{n} \mathsf{b}^{(i)} \log \frac{p_i}{1 - p_i} + \log(1 - p_i)$$

Since $p_i$ are given, the maximization problem turns into

$$\arg\max_{\mathsf{b}^{(1)}, \ldots, \mathsf{b}^{(n)}} \sum_{i=1}^{n} \mathsf{b}^{(i)} \log \frac{p_i}{1 - p_i}$$

Obviously, if without the constraint, we should set all the $\mathsf{b}^{(i)}$ with $\log \frac{p_i}{1-p_i} > 0$ to 1. That is, select the variables with Bernoulli parameters greater than 0.5. With the constraints, since $\frac{p_i}{1-p_i}$ grows monotonically with respect to $p_i$, we should first sort $p_i$s, and set $\mathsf{b}^{(i)}$ to 1 for the top $B$ $p_i$ satisfying $\log \frac{p_i}{1-p_i} > 0$. This results in Algorithm 3.

A.4 Proof of Theorem 4.3

Theorem 4.3 claims that FCNN has full expressivity for function-free Horn rules. [9] Formally, given a finite set of Horn rules $\mathcal{R}$, there exists a FCNN model, parameterized by $\boldsymbol{\theta}$, from which we can extract $\mathcal{R}$ according to Section 4.4.

As preparation, we first set the hyperparameters and relevant concepts. Let the number of universal meta-rules equal to the number of Horn rules, i.e., $|\mathcal{U}| = |\mathcal{R}|$. Let $V$ equal to the maximum number of variables for any rule in $\mathcal{R}$, $B$ equal to the maximum number of body atoms for any rule in $\mathcal{R}$. Let $\mathcal{P}$ be all the predicates appearing in $\mathcal{R}$. Let $M$, the multiplicity of body atoms, be the maximum number of times a predicate appears in the body of any single rule in $\mathcal{R}$. Let $A$ be the maximum arity for all the predicates in $\mathcal{P}$. In addition, let $J$ denote the maximum number of arguments for any rule in $\mathcal{R}$. As in Sec. 4.1, let $\mathcal{A}^{\mathrm{h}}(\mathcal{P}) := \{p(\arg_1^p, \ldots, \arg_{\mathrm{arity}(p)}^p) \,|\, p \in \mathcal{P}\}$ be the set of head-candidate atoms, $\mathcal{A}^{\mathrm{b}}(\mathcal{P}, M) := \{p(\arg_1^{p,m}, \ldots, \arg_{\mathrm{arity}(p)}^{p,m}) \,|\, p \in \mathcal{P}, m \in [M]\}$ be the set of body-candidate atoms.[10]

In the following, we are going to construct $\boldsymbol{\theta}$ based on $\mathcal{R}$. Intuitively, since each given rule has deterministic atoms and variables, we simply set $\boldsymbol{\theta}$ to let all the unification probability values corresponding to this rule's atoms and variables be the maximums. Observing that the model parameters are meta-rule-independent, except for predicates, our constructive proof below avoids the effects of predicate embeddings so that we can capture each given rule by one meta-rule. Hence, in the following, we consider a given rule in $\mathcal{R}$, and use one meta-rule to model it.

The following proof starts by unifying variables with arguments and finish by unifying atoms.

**Head variable unification** Consider a given Horn rule in $\mathcal{R}$, let $p$ denote the head atom's predicate, $i$ denote the head atom's $i$-th argument, and $v^*$ denote the index of the logical variable this argument takes[11]. Besides, let $j$ denote the number index of that argument among all the atoms' arguments in the given rule (including body atoms' arguments)[12]. By definition, there exists exactly one element in $\mathcal{A}^{\mathrm{h}}(\mathcal{P})$ related to $p$. Let $a$ denote this element. Our goal is to ensure

$$v^* = \underset{v \in [V]}{\arg\max} \, P_{\boldsymbol{\theta}}(\mathrm{v}_{a,i}^{\mathrm{h}} = v)$$

by setting appropriate embeddings. As in the main text, we omit the meta-rule index here. Recall that

$$P_{\boldsymbol{\theta}}(\mathrm{v}_{a,i}^{\mathrm{h}} = v) = \mathrm{softmax}\left(\boldsymbol{\eta}_{a,i}^{\mathrm{h}} \cdot \boldsymbol{\nu}_1, \ldots, \boldsymbol{\eta}_{a,i}^{\mathrm{h}} \cdot \boldsymbol{\nu}_V\right)_v.$$

Therefore, it is sufficient to let $\boldsymbol{\eta}_{a,i}^{\mathrm{h}} \cdot \boldsymbol{\nu}_{v^*} = 1$ and $\boldsymbol{\eta}_{a,i}^{\mathrm{h}} \cdot \boldsymbol{\nu}_v = 0$ for $v \neq v^*$. To this end, we set $d_{\mathrm{v}} = J+1$ (dimension of variable embeddings and argument embeddings), set variable embeddings

$$\boldsymbol{\nu}_{v^*,j} = 1$$
$$\boldsymbol{\nu}_{v,j} = 0, \quad \forall v \neq v^*$$
$$\boldsymbol{\nu}_{v,J+1} = 0, \quad \forall v \in [V],$$

and set the argument embeddings

$$\boldsymbol{\eta}_{a,i,j}^{\mathrm{h}} = 1$$
$$\boldsymbol{\eta}_{a,i,j'}^{\mathrm{h}} = 0, \quad \forall j' \in [J], j' \neq j.$$

The new index appended to the subscript of each embedding indicates the elements. The element index starts from 1. The above setup has determined the first $J$ elements of the argument embedding $\boldsymbol{\eta}_{a,i}^{\mathrm{h}}$ and the $j$-th and $(J+1)$-th element of all the variable embeddings $\boldsymbol{\nu}_1, \ldots, \boldsymbol{\nu}_V$, and has ensured $\boldsymbol{\eta}_{a,i}^{\mathrm{h}} \cdot \boldsymbol{\nu}_{v^*} = 1$ and $\boldsymbol{\eta}_{a,i}^{\mathrm{h}} \cdot \boldsymbol{\nu}_v = 0$ for $v \neq v^*$. After setting the variable embeddings for other

---

[9]Without loss of forward chaining results, functions can be replaced with predicates.

[10]If $\mathcal{R}$ is a solution of an ILP task, we may set $\mathcal{P}^{\mathrm{bk}}, \mathcal{P}^{\mathrm{aux}}, \mathcal{P}^{\mathrm{tgt}}$ separately and have $\mathcal{P} := \mathcal{P}^{\mathrm{bk}} \cup \mathcal{P}^{\mathrm{aux}} \cup \mathcal{P}^{\mathrm{tgt}}$, as in Sec. 4.1. The following proof holds for any given $\mathcal{P}$. In CWA mode, the head-candidate atoms are limited to $\mathcal{P}^{\mathrm{aux}} \cup \mathcal{P}^{\mathrm{tgt}}$, and the following proof still holds.

[11]The variable indexes can be arbitrarily set.

[12]The argument indexes can be arbitrarily set.

arguments in the same way (including body atoms' arguments, see below), the variable embeddings are also fully determined [13]. The $(J+1)$-th element of the argument embeddings are reserved for atom unification and will be determined in atom unification. Consequently, setting the $(J+1)$-th element of the variable embeddings to 0 avoids the influence of the $(J+1)$-th element of the argument embeddings.

**Body variable unification** Setting variable embeddings and argument embeddings according to body variable unification is almost the same as head variable unification. While the head variable unification only considers one atom, the head atom, body variable unification considers every body atom in the same way. The critical difference arises from $\mathcal{A}^{\mathrm{b}}(\mathcal{P}, M)$. Different from $\mathcal{A}^{\mathrm{h}}(\mathcal{P})$, there exists multiple elements in $\mathcal{A}^{\mathrm{b}}(\mathcal{P}, M)$ related to $p$ if $M > 1$. To deterministically select an element from $\mathcal{A}^{\mathrm{b}}(\mathcal{P}, M)$, we can index the occurrence of $p$ in the body of the given rule [14]. Considering one of its occurrence, say, the $m$-th occurrence, we select $a := p(\arg_1^{p,m}, \ldots, \arg_{\mathrm{arity}(p)}^{p,m})$ from $\mathcal{A}^{\mathrm{b}}(\mathcal{P}, M)$. Then, every thing is the same as in head variable unification.

**Head atom unification** Again, consider a given Horn rule in $\mathcal{R}$, let $p$ denote the head atom's predicate. By definition, there exists exactly one element in $\mathcal{A}^{\mathrm{h}}(\mathcal{P})$ related to $p$. Let $a^*$ denote this element. Our goal is to ensure

$$a^* = \arg\max_{a \in \mathcal{A}^{\mathrm{h}}} P_{\boldsymbol{\theta}}(\mathsf{a}^{\mathrm{h}} = a)$$

by setting appropriate embeddings. Again, we omit the meta-rule index here. Recall that

$$P_{\boldsymbol{\theta}}(\mathsf{a}^{\mathrm{h}} = a) = \mathrm{softmax}\left(\boldsymbol{\eta}_a^{\mathrm{h}} \cdot \boldsymbol{\nu}^{\mathrm{h}} : a \in \mathcal{A}^{\mathrm{h}}(\mathcal{P})\right)_a$$

and

$$\boldsymbol{\eta}_a^{\mathrm{h}} := \boldsymbol{W}^{\mathrm{h}} \cdot [\boldsymbol{\eta}_p, \boldsymbol{\eta}_{a,1}^{\mathrm{h}}, \ldots, \boldsymbol{\eta}_{a,A}^{\mathrm{h}}] \in \mathbb{R}^{d_{\mathrm{a}}}.$$

Therefore, it is sufficient to let $\boldsymbol{\eta}_{a*}^{\mathrm{h}} \cdot \boldsymbol{\nu}^{\mathrm{h}} = 1$ and $\boldsymbol{\eta}_a^{\mathrm{h}} \cdot \boldsymbol{\nu}^{\mathrm{h}} = 0$ for $a \neq a^*$. For simplicity, we set $d_{\mathrm{a}} = d_{\mathrm{p}} + A \cdot d_{\mathrm{v}}$ and set $\boldsymbol{W}^{\mathrm{h}}$ to be the identity matrix. We set the embedding of HEAD

$$\boldsymbol{\nu}_{d_{\mathrm{p}}+J+1}^{\mathrm{h}} = 1$$
$$\boldsymbol{\nu}_{j'}^{\mathrm{h}} = 0, \quad \forall j' \neq d_{\mathrm{p}} + J + 1$$

and set argument embeddings

$$\boldsymbol{\eta}_{a^*,1,J+1}^{\mathrm{h}} = 1$$
$$\boldsymbol{\eta}_{a,1,J+1}^{\mathrm{h}} = 0, \quad \forall a \neq a^*$$
$$\boldsymbol{\eta}_{a,k,J+1}^{\mathrm{h}} = 0, \quad \forall a \in \mathcal{A}^{\mathrm{h}}(\mathcal{P}), k \in \{2, \ldots, A\}$$

As mentioned before, the $(J+1)$-th element of argument embeddings are reserved for atom unification, and only the first argument plays the important role here. The above setup ensures $\boldsymbol{\eta}_{a^*}^{\mathrm{h}} \cdot \boldsymbol{\nu}^{\mathrm{h}} = 1$ and $\boldsymbol{\eta}_a^{\mathrm{h}} \cdot \boldsymbol{\nu}^{\mathrm{h}} = 0$ for $a \neq a^*$. As mentioned before, we avoid the effect from predicate embeddings and they can be set to any values.

**Body atom unification** Again, consider a given Horn rule in $\mathcal{R}$. As mentioned in body variable unification, we can select exactly one element from $\mathcal{A}^{\mathrm{b}}(\mathcal{P}, M)$ for each body atom of the given rule. Let $\mathcal{A}$ denote the set of all these elements for the rule. According to Algorithm 3, our goal is to ensure

$$P_{\boldsymbol{\theta}}(\mathsf{a}_a^{\mathrm{b}} = 1) > 0.5, \quad \forall a \in \mathcal{A}$$
$$P_{\boldsymbol{\theta}}(\mathsf{a}_a^{\mathrm{b}} = 1) \leq 0.5, \quad \forall a \notin \mathcal{A}$$

by setting appropriate embeddings. Again, we omit the meta-rule index here. Recall that

$$P_{\boldsymbol{\theta}}(\mathsf{a}_a^{\mathrm{b}} = 1) := \sigma\left(\boldsymbol{\eta}_a^{\mathrm{b}} \cdot \boldsymbol{\nu}^{\mathrm{b}}\right), \forall a \in \mathcal{A}^{\mathrm{b}}(\mathcal{P}, M)$$

---

[13]Some rules may have fewer than $J$ arguments. Simply set the redundant elements of variables embeddings to 0.

[14]The occurrence indexes can be arbitrarily set.

and

$$\boldsymbol{\eta}_a^{\mathrm{b}} := \boldsymbol{W}^{\mathrm{b}} \cdot [\boldsymbol{\eta}_p, \boldsymbol{\eta}_{a,1}^{\mathrm{b}}, \dots, \boldsymbol{\eta}_{a,A}^{\mathrm{b}}] \in \mathbb{R}^{d_{\mathrm{a}}}$$

Therefore, it is sufficient to let $\boldsymbol{\eta}_a^{\mathrm{b}} \cdot \boldsymbol{\nu}^{\mathrm{b}} = 1$ for $a \in \mathcal{A}$ and $\boldsymbol{\eta}_a^{\mathrm{b}} \cdot \boldsymbol{\nu}^{\mathrm{b}} = 0$ for $a \notin \mathcal{A}$. Recall that we have set $d_{\mathrm{a}} = d_{\mathrm{p}} + A \cdot d_{\mathrm{v}}$. Similar to $\boldsymbol{W}^{\mathrm{h}}$, we set $\boldsymbol{W}^{\mathrm{b}}$ to be the identity matrix. Similar to head atom unification, we set the embedding of BODY

$$\boldsymbol{\nu}_{d_{\mathrm{p}}+J+1}^{\mathrm{b}} = 1$$
$$\boldsymbol{\nu}_{j'}^{\mathrm{b}} = 0, \quad \forall j' \neq d_{\mathrm{p}} + J + 1$$

and set argument embeddings

$$\boldsymbol{\eta}_{a,1,J+1}^{\mathrm{b}} = 1, \quad \forall a \in \mathcal{A}$$
$$\boldsymbol{\eta}_{a,1,J+1}^{\mathrm{b}} = 0, \quad \forall a \notin \mathcal{A}$$
$$\boldsymbol{\eta}_{a,k,J+1}^{\mathrm{b}} = 0, \quad \forall a \in \mathcal{A}^{\mathrm{h}}(\mathcal{P}), k \in \{2, \dots, A\}$$

# B  IMPLEMENTATION DETAILS

In this section, we introduce the implementation details of FCNN and LLM-based ILP.

## B.1  FCNN DETAILS

We use Adam optimizer (Kingma, 2014) to optimize the objective (Eq. 6) by computing gradient according to Eq. 7.

Hyperparameters and more implementation details, including forward chaining and evaluation, can be found in our released code.

## B.2  LLM DETAILS

The prompt used for ILP tasks is given below

```
Task description: given the following facts and positive target
examples, induce a set of Horn rules that, together with the
facts, can entail all the positive target examples and not
others. The facts and positive target examples are all Horn
facts.

Facts:
{FACTS}

Positive target examples:
{POSITIVE_TARGET_EXAMPLES}

You must format your rules as follows (variables must be capital
letters and predicates must be lowercase letters):
rmwq(X) :- coa(Y), dska(Y,Z), dska(Z,W).
ovm(Y,X) :- mbjd(Z), wquv(Z,X,Y).
...
```

{FACTS} is comma separated Horn facts and {POSITIVE_TARGET_EXAMPLES} is comma separated positive target examples.

The predicates and constants in the facts and positive target examples are converted to random strings. This follows the common practice of ILP, i.e., do not tell an ILP system the meaning of predicates and constants, which should be treated as meaningless words. Such a setting requires ILP systems to learn the semantics of the symbols (i.e., predicates, constants) completely from data.

Table 5: Full results on 20 small-scale ILP tasks

| | Task | $\partial$ILP | LRI | HRI | DFORL | FCNN |
|---|---|---|---|---|---|---|
| | Predecessor | 100 | 100 | 100 | 100 | 100 |
| | Even-Odd | 100 | 100 | 40 | 100 | 100 |
| Arithmetic | Even-Succ2 | 48.5 | 100 | 40 | 100 | 100 |
| | LessThan | 100 | 100 | 100 | 100 | 100 |
| | Fizz | 10 | 10 | 0 | 0 | 100 |
| | Buzz | 35 | 70 | 40 | 0 | 100 |
| Lists | Length | 92.5 | 100 | 0 | 100 | 100 |
| | Member | 100 | 100 | 100 | 100 | 100 |
| | Son | 100 | 100 | 100 | 100 | 100 |
| | Grandparent | 96.5 | 100 | 100 | 100 | 100 |
| Family Tree | Husband | 100 | - | - | 100 | 100 |
| | Uncle | 70 | - | - | 100 | 100 |
| | Relatedness | 100 | 100 | 100 | 100 | 100 |
| | Father | 100 | 100 | - | 100 | 100 |
| | Directed Edge | 100 | 100 | 100 | 100 | 100 |
| | Adjacent to Red | 50.5 | 100 | 100 | 100 | 100 |
| Graph | Two Children | 95 | 0 | 100 | 100 | 100 |
| | Graph Coloring | 94.5 | 0 | 100 | 100 | 100 |
| | Connectedness | 100 | 100 | 100 | 100 | 100 |
| | Cyclic | 100 | 100 | 100 | 100 | 100 |

Table 6: Full comparison of CWA and OWA mode

| Task | CWA | OWA |
|---|---|---|
| circle_1 | **80.0 ± 0.0** | 79.3 ± 2.2 |
| circle_2 | **80.1 ± 2.7** | 79.0 ± 2.2 |
| cong_1 | 81.3 ± 2.8 | **81.8 ± 0.6** |
| para_1 | **96.5 ± 2.9** | 93.8 ± 6.4 |
| para_2 | **83.0 ± 4.2** | 82.3 ± 2.0 |
| para_3 | **98.5 ± 1.2** | 97.8 ± 1.5 |
| perp_1 | 83.4 ± 0.0 | **84.1 ± 2.2** |
| eqangle_1 | **72.5 ± 6.2** | 70.4 ± 3.4 |
| eqangle_2 | 67.1 ± 4.4 | **69.4 ± 5.0** |
| eqratio_1 | **68.0 ± 8.8** | 63.3 ± 4.2 |
| eqratio_2 | **67.6 ± 5.5** | 64.4 ± 9.2 |
| eqratio_3 | **66.1 ± 3.8** | 63.5 ± 4.8 |
| Avg. | **78.7** | 77.4 |

## C    EXPERIMENTAL RESULTS

### C.1    FULL RESULTS ON 20 SMALL-SCALE TASKS

Table 5 lists all results on the 20 small-scale ILP tasks.

### C.2    COMPARISON BETWEEN CWA AND OWA

Table 6 show the full comparison between FCNN's CWA mode and OWA mode on GeoILP. The OWA mode shows slightly weaker performance, probably since the OWA mode suffers from a larger hypothesis space.

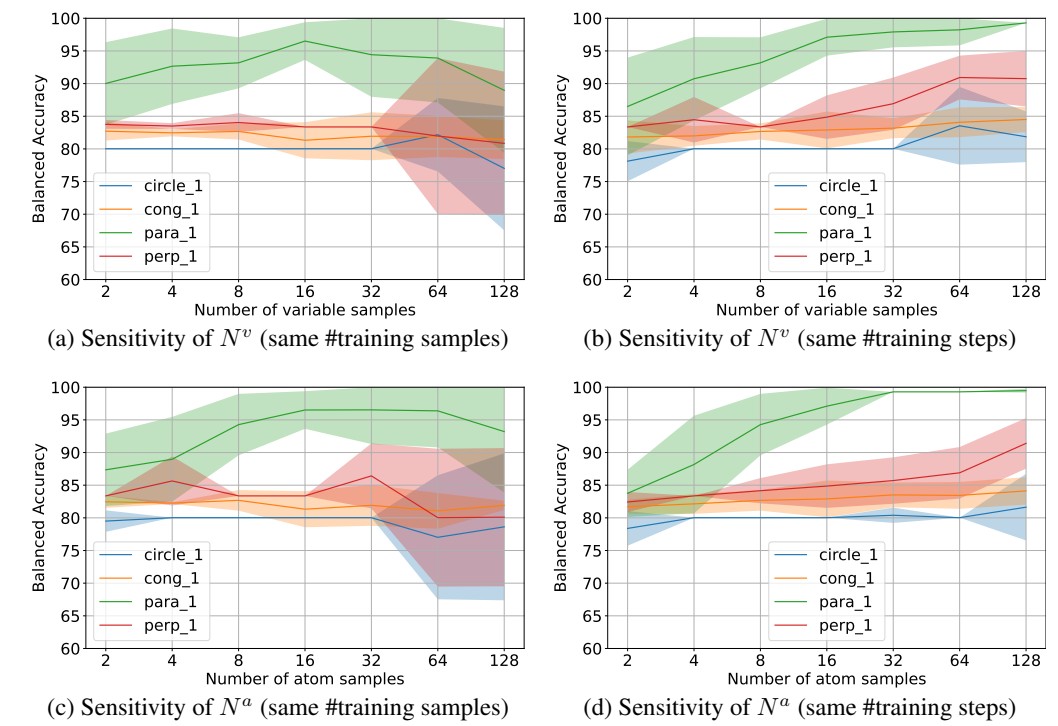

(a) Sensitivity of $N^v$ (same #training samples)

(b) Sensitivity of $N^v$ (same #training steps)

(c) Sensitivity of $N^a$ (same #training samples)

(d) Sensitivity of $N^a$ (same #training steps)

Figure 1: Sensitivity of sample size

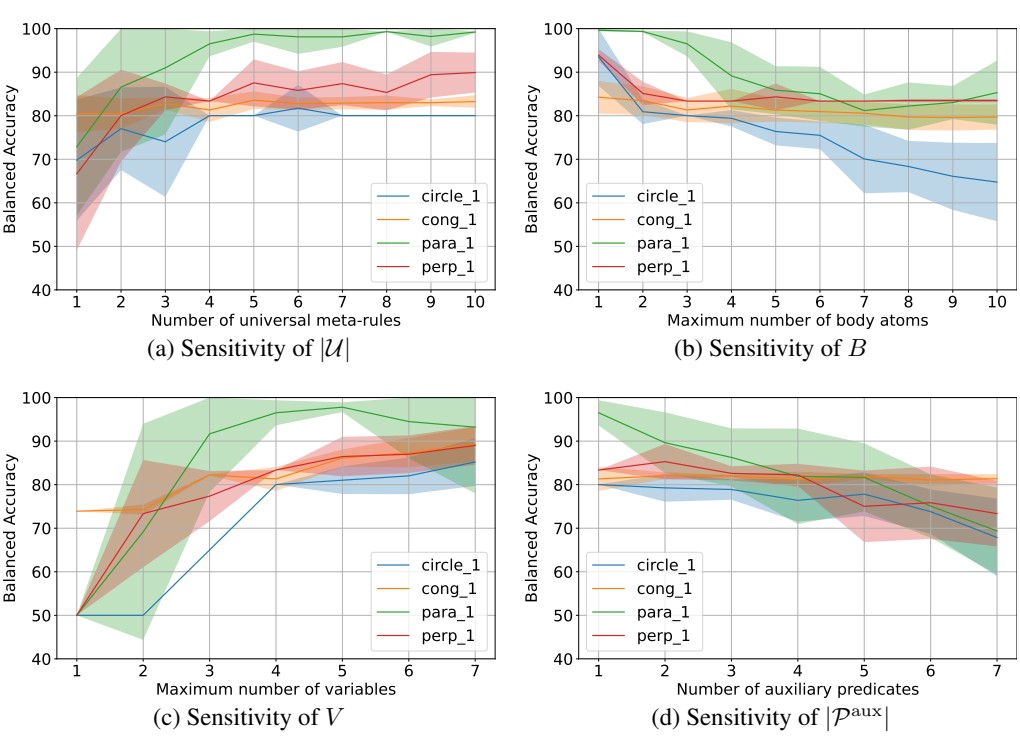

(a) Sensitivity of $|\mathcal{U}|$

(b) Sensitivity of $B$

(c) Sensitivity of $V$

(d) Sensitivity of $|\mathcal{P}^{\mathrm{aux}}|$

Figure 2: Sensitivity of symbolic hyperparameters

Table 7: Comparison of schedulers for entropy regularization (the last row refers to exponential decay)

| Scheduler | circle_1 | cong_1 | para_1 | perp_1 |
|---|---|---|---|---|
| $\lambda = 0$ | $50.5 \pm 1.6$ | $77.7 \pm 3.5$ | $59.0 \pm 11.6$ | $56.7 \pm 9.7$ |
| $\lambda = 10^{-2}$ | $\mathbf{80.0 \pm 0.0}$ | $\mathbf{84.4 \pm 2.8}$ | $96.3 \pm 3.2$ | $\mathbf{83.4 \pm 0.0}$ |
| $\lambda = 10^{-6}$ | $64.7 \pm 12.8$ | $79.5 \pm 2.4$ | $73.2 \pm 13.6$ | $69.4 \pm 10.5$ |
| $\lambda = 10^{-2}$ to $\lambda = 10^{-6}$ | $\mathbf{80.0 \pm 0.0}$ | $81.3 \pm 2.8$ | $\mathbf{96.5 \pm 2.9}$ | $\mathbf{83.4 \pm 0.0}$ |

Table 8: Time usage (in seconds)

| Task | HRI | DFORL | FCNN | LLM |
|---|---|---|---|---|
| Predecessor | $4 \pm 0$ | $115 \pm 50$ | $2 \pm 0$ | $64 \pm 20$ |
| LessThan | $20 \pm 0$ | $126 \pm 106$ | $20 \pm 17$ | $73 \pm 17$ |
| Cyclic | $69 \pm 1$ | $326 \pm 163$ | $148 \pm 123$ | $104 \pm 11$ |
| Relatedness | $122 \pm 40$ | $372 \pm 83$ | $63 \pm 51$ | $135 \pm 38$ |

## C.3 HYPERPARAMETER ANALYSIS

Figure 2 analyzes the sensitivity of the number of universal meta-rules $|\mathcal{U}|$, maximum number of body atoms $B$, maximum number of variables $V$, and the number of auxiliary predicates $|\mathcal{P}^{\mathrm{aux}}|$. On one hand, increasing the number of universal meta-rules and maximum number of variables can generally improve the performance. Before a threshold (say, 5 for 2.a and 4 for 2.c), the performance increases faster, since the model has insufficient capacity in this region. After the threshold, the performance gain becomes smaller, oscillating, or even negative (rare). On the other hand, increasing the maximum number of body atoms and the number of auxiliary predicates causes severe performance decrease. Considering that the hypothesis space increases when these two hyperparameters increase quickly, we argue that the performance decrease is due to insufficient optimization. The opposite performance of the two groups of hyperparameters is due to their different impacts. For example, while adding a body atom typically makes a useful Horn rule useless (i.e., not produce new facts during forward chaining since the body cannot match any existing facts), adding a new universal meta-rule may increase the possibility of successfully sampling correct rules.

Figure 3 analyzes the sensitivity of predicate embedding dimension $d_{\mathrm{p}}$, variable embedding dimension $d_{\mathrm{v}}$, and atom embedding dimension $d_{\mathrm{a}}$. In general, the performance decreases and becomes unstable when any dimension increases. We argue that this is caused by the increased optimization difficulty. This performance drop is significantly worse for the atom embedding dimension. A small atom embedding dimension may effectively compress information for atoms; however, a large atom embedding dimension may increase optimization difficulty.

Table 7 compares different schedulers for the entropy regularization. Results show that sufficient (i.e. high $\lambda$) entropy regularization is critical for good performance. Besides, the exponential decay of $\lambda$ yields different performance on different tasks, and achieves comparable performance with a fixed and high $\lambda$.

## C.4 COMPARISON OF TIME USAGE

Table 8 compares the time usage among HRI, DFORL, FCNN on 1 CPU & 1 GPU. The selected tasks have varying difficulties in terms of program size, i.e., the total number of atoms in the solution provided in Evans & Grefenstette (2018). We also add LLM's time usage, but it is not comparable to others since we call the public API, and the resource behind the API is unknown (probably significantly more expensive than the resource we use).

The results reveal that DFORL is the slowest, probably because DFORL requires an expensive preprocessing for constant substitution. FCNN and HRI take about the same amount of time on

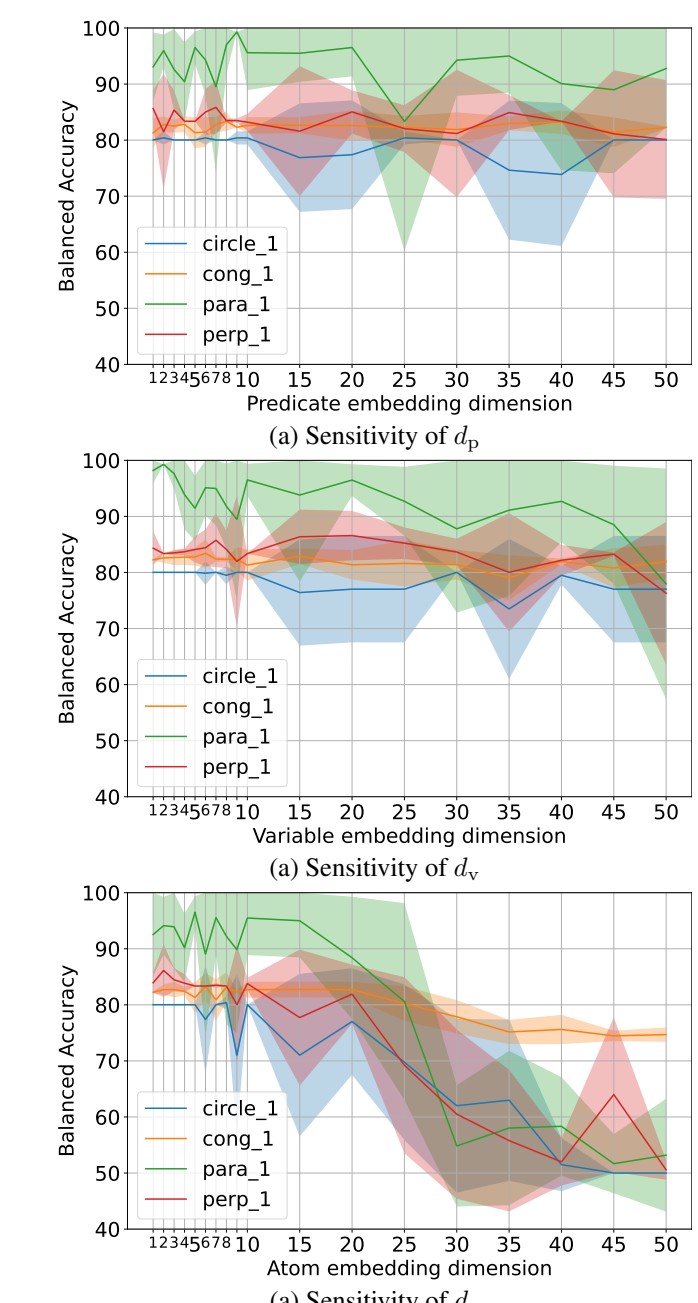

Figure 3: Sensitivity of embedding dimensions

average, justifying that, although FCNN supports richer language bias, FCNN's time consumption is comparable to existing works.

In addition, we study how runtime scales with respect to the maximum number of body atoms $B$ and the number of predicates. Since the number of background predicates and target predicates is fixed for a given ILP task, we vary the number of predicates by varying the number of auxiliary predicates $|\mathcal{P}^{\mathrm{aux}}|$. Table 4 shows that the runtime does not increase significantly due to these two factors, demonstrating FCNN's scalability. The high variance on para_1 with small $B$ is due to early stopping, which will be triggered if the model achieves 100% balanced accuracy on training data.

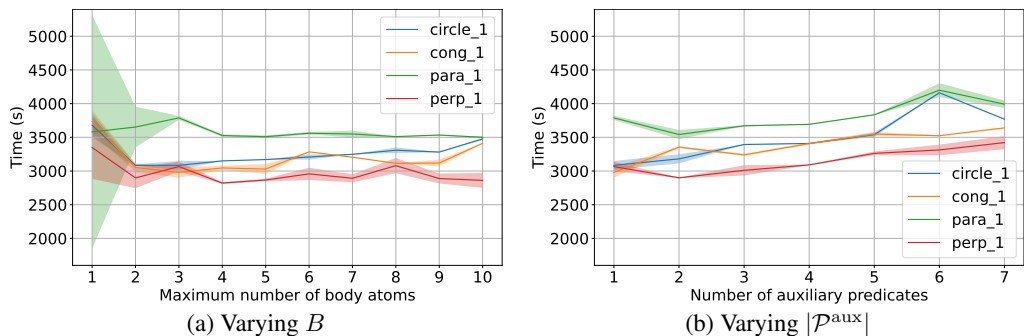

(a) Varying $B$        (b) Varying $|\mathcal{P}^{\mathrm{aux}}|$

Figure 4: Runtime curves (same maximum training iterations)

## D    LLM USAGE

We utilize large language models for grammatical correction. In addition, we employ large language models to check whether the Algorithm 2 has been proposed in existing literature.

