# OpenReview forum: "Forward Chaining Neural Network for Rule Induction"
_ICLR.cc/2026/Conference — Submitted to ICLR 2026_

### Official Review · Reviewer_t1ZP · 2025-10-30

**Soundness:** 3
**Presentation:** 2
**Contribution:** 3
**Rating:** 6
**Confidence:** 2

**Summary:**

This paper proposes the Forward Chaining Neural Network (FCNN), a neuro-symbolic model for Inductive Logic Programming (ILP). Unlike earlier neural ILP approaches (e.g., ∂ILP, LRI, HRI, DFORL), FCNN introduces a universal meta-rule framework that allows learning Horn rules of arbitrary arity and body length, under both closed-world and open-world assumptions. The method relaxes symbolic unification into a continuous probabilistic framework, parameterizing head and body atoms with embeddings and Bernoulli random variables. Optimization is performed using nested REINFORCE estimators and entropy regularization. Experiments on both classic small-scale ILP benchmarks and the new large-scale GeoILP dataset show that FCNN outperforms previous neuro-symbolic systems and even recent reasoning LLMs on most tasks.

**Strengths:**

The idea of fully relaxing Horn rule unification into a differentiable stochastic process is elegant and theoretically sound. The probabilistic modeling of atom and variable unification with Bernoulli and categorical distributions provides a flexible parameterization.

The paper is rigorous, with clear mathematical definitions, probabilistic modeling, and proofs (e.g., unbiased gradient estimator, completeness theorem). Algorithmic details for differentiable subset sampling are also well described.

The paper provides a concrete procedure (Algorithm 2) for extracting interpretable symbolic rules from the learned stochastic representations — an important feature for ILP research.

**Weaknesses:**

Presentation and readability.

Lack of conceptual comparisons.

Limited ablations on modeling choices.


i will detail these points in the section below.

**Questions:**

**Presentation and readability.** The paper’s exposition is overly dense and bottom-up. The intuition behind the model (e.g. idea of reparametrizing unifications) could have benefited from a clearer top-down narrative — motivating ideas first, then details. A graphical illustration could also have helped.

**Limited ablations on modeling choices.** Although prior neuro-symbolic systems like ∂ILP, LRI, HRI, and DFORL are discussed, comparisons to alternative paradigms that feel closer in paradigm misses. For example, DiffLog [1] and AlphaILP[2] exploited templates and forward chaining for ILP. How does it differ? Also, DeepSoftLog[3] introduces the idea of soft unification into probabilisti logic programming. Soft unification is actually a different parameterization (kernel-like) of a probabilisticu unification. Also in that paper there was some ILP on automata. What are the links?

**Limited ablations on modeling choices.** The ablations mainly address sample sizes and OWA vs CWA modes. Missing are experiments probing the necessity of specific design choices (e.g., embeddings vs discrete parameters, REINFORCE vs relaxation-based training, as many other models do).


[1] Si, Xujie, et al. "Synthesizing datalog programs using numerical relaxation." arXiv preprint arXiv:1906.00163 (2019).
[2] Shindo, Hikaru, et al. "Learning differentiable logic programs for abstract visual reasoning." Machine Learning 113.11 (2024): 8533-8584.
[3] Maene, Jaron, and Luc De Raedt. "Soft-unification in deep probabilistic logic." Advances in Neural Information Processing Systems 36 (2023): 60804-60820.

---

> ### Author Response · Authors · 2025-11-25
>
> We sincerely thank you for the insightful comments and constructive suggestions, which are highly valuable in improving our manuscript.
> We also appreciate your recognition of our method's elegance and theoretical soundness.
>
> In the following, we address your concerns point by point and detail how we revised the paper based on your suggestions.
> **(The reference number or cited texts correspond to the revised paper, which we have already uploaded to the review system.)**
>
>
>
> > Weakness 1 \& Question 1: Presentation and readability. The paper’s exposition is overly dense and bottom-up. The intuition behind the model (e.g. idea of reparametrizing unifications) could have benefited from a clearer top-down narrative — motivating ideas first, then details. A graphical illustration could also have helped.
>
> According to your suggestions, we revised Section 4 (the method section).
> Especially, we enrich the first three paragraphs of Section 4 and the first paragraph of Section 4.2 to explain the intuition behind the model in a top-down manner. We invite you to read our revised manuscript to see whether it meets your suggestions.
>
> In addition, we add a pseudo-code (Algorithm 1) to illustrate the overall learning procedure, which also contribute to illustrate our method from a high level. We also try to draw a illustration figure as you suggested, but we found the figure is intricate and the pseudo-code is easier for understanding. We are willing to add a figure if you found a figure is still necessary.
>
>
>
> > Weakness 2 \& Question 2: Lack of conceptual comparisons. Although prior neuro-symbolic systems like ∂ILP, LRI, HRI, and DFORL are discussed, comparisons to alternative paradigms that feel closer in paradigm misses. For example, DiffLog [1] and AlphaILP[2] exploited templates and forward chaining for ILP. How does it differ? Also, DeepSoftLog[3] introduces the idea of soft unification into probabilisti logic programming. Soft unification is actually a different parameterization (kernel-like) of a probabilisticu unification. Also in that paper there was some ILP on automata. What are the links?
>
> According to your provided references, we find several more related works and add discussions for all of them in Section 2 Related Work.
> Below is a discussion for newly added references.
> - Difflog (Si et al., 2019) leverages a rule-level relaxation manner (similar to $\partial$ ILP) and requires a rule-candidate set, also restricted to limited language bias and intractable for large-scale ILP tasks in terms of memory consumption (Chen et al., 2025). Similar rule-level relaxations have been explored in $\partial$ ILP-ST (Shindo et al., 2021), $\alpha$ ILP (Shindo et al., 2023), and NEUMANN (Shindo et al., 2024), but require high-quality hypothesis space to obtain good performance (Shindo et al., 2021).
>     - It is worth mentioning that $\alpha$ ILP and NEUMANN are mainly for combining visual perception and logical models. Their ILP components are simply extensions from $\partial$ ILP-ST and thus inherit its shortcomings (i.e., require high-quality hypothesis space to obtain good performance).
> - Neural Theorem Prover (Rockt¨aschel & Riedel, 2017)) and DeepSoftLog (Maene & Raedt, 2023)) are designed for logic programming. Although they supports inductive logic programming by predicate-level relaxations (e.g., the automata experiments in DeepSoftLog), they also require human-defined task-specific templates.
>     - To realize soft unification, DeepSoftLog uses a kernel according to the distance on a hypersphere. The predicate unification in Neural Theorem Prover is also kernel-based. LRI, which also incorperates predicate-level unifications, uses cosine similarity to do the unification. Our FCNN does not directly unify predicates. FCNN's unifications are based on inner product.
>
>
>
> > Weakness 3 \& Question 3: Limited ablations on modeling choices. The ablations mainly address sample sizes and OWA vs CWA modes. Missing are experiments probing the necessity of specific design choices (e.g., embeddings vs discrete parameters, REINFORCE vs relaxation-based training, as many other models do).
>
> Sorry, we didn't quite understand this question and need your help.
> - What is discrete parameters? To the best of our knowledge, all neuro-symbolic ILP use real-value model parameters. And we did not see any possibility for integrating discrete parameters in FCNN.
> - Our training IS a relaxation-based training, which relaxes Horn rules into a continuous space. To optimize in this relaxed space, we use a REINFORCE-like gradient estimator.
>
> In Appendix C.3, according to other reviewers' suggestions, we further analyze the effect of several hyperparameters, including the number of meta-rules, maximum number of body atoms, maximum number of variables, number of predicates, embedding dimensions, entropy scheduler.
>
>
>
>
>
>
> Thank you again for the thorough review. We would be glad to provide any additional information if needed.

---

### Official Review · Reviewer_CJSz · 2025-10-30

**Soundness:** 3
**Presentation:** 3
**Contribution:** 2
**Rating:** 4
**Confidence:** 4

**Summary:**

This submission investigates neural-symbolic inductive logic programming (ILP) with the goal of learning logical rules from data. The core mechanism employed is a forward-chaining neural network based on so-called meta-rules (in the form of Horn clauses). Its main advantage lies in relaxing syntactic constraints on rules—for example, allowing the arity of predicates to increase—and thereby supporting, to some extent, the open-world assumption.

**Strengths:**

- The paper demonstrates some progress compared to prior work. In particular, while previous approaches typically impose strong restrictions on atoms, the current method can, in principle, handle general Horn clauses.

- The paper is written in a formal and structured manner, which contributes to its clarity and readability.

**Weaknesses:**

-  Limited novelty. The work follows a fairly standard approach to tackling ILP in a neuro-symbolic manner—namely, by neuralizing logic rules (in this case, via forward-chaining neural networks) and softening symbols through distributions, thereby effectively continuizing traditional discrete objects. The use of REINFORCE is also conventional. In this respect, the contribution appears to lie primarily in adapting existing techniques to the ILP setting, rather than introducing fundamentally new ideas.

-  Overstated contributions. The framework remains template-based, albeit with a more relaxed form than prior work. Consequently, its ability to address the open-world assumption (OWA) is still limited. For example, it is unclear whether the proposed method can discover new predicates or formulate new symbolic concepts, rather than relying on predefined templates.

- Limited impact. The experimental evaluation is restricted to ILP benchmark datasets, which are relatively small and arguably toy problems. Given that ILP represents a niche research area, the scope of the current paper appears narrow, as reflected in the limited breadth of related work. It remains uncertain—though IMHO unlikely—whether the proposed method can generalize to real-world applications.

**Questions:**

- Line 059, without **strong** language bias. What does strong mean here?
- Line 205-206, I do not really understand “ i.e., unifying the head atom according to a probability”, “Such a unification corresponds to the property that the Horn rule only allows one head atom.”
- Line 295 “After optimizing, the distributions are supposed to collapse to deterministic distributions, […]” why?

---

> ### Author Response · Authors · 2025-11-25
>
> We sincerely thank you for the insightful comments and constructive suggestions, which are highly valuable in improving our manuscript.
> We also appreciate your recognition of the meaningful progress made by our method and the clarity & readability of our manuscript.
>
> In the following, we address your concerns point by point and detail how we revised the paper based on your suggestions.
> **(The reference number or cited texts correspond to the revised paper, which we have already uploaded to the review system.)**
>
>
>
> > Weakness 1: Limited novelty. The work follows a fairly standard approach to tackling ILP in a neuro-symbolic manner—namely, by neuralizing logic rules (in this case, via forward-chaining neural networks) and softening symbols through distributions, thereby effectively continuizing traditional discrete objects. The use of REINFORCE is also conventional. In this respect, the contribution appears to lie primarily in adapting existing techniques to the ILP setting, rather than introducing fundamentally new ideas.
>
> - The fundamental idea of FCNN do follows the mainstream of neuralizing logic rules. However, **the existing neuralizing approaches fail to extend to richer expressiveness** (i.e., fail to support richer language biases).
>     - Take HRI as an example. HRI proposes a model that can capture a limited set of Horn rules, e.g., arity $\leq$ 2, less than two body atoms. However, **the model can not be extended to higher arity or more body atoms**. At least, HRI's paper did not show how to design such model for richer expressiveness and we failed to find a way to extend it.
>     - Hence, to support any Horn rule, we propose to directly learn logical variables unification and atom unification. Note that, regarding unification learning, existing works only learn predicate unification, which requires predefined variables, i.e., requires specially designed templates. However, **our proposed model do not need such special templates** (See our response to *Weakness 2*).
>     - However, learning the unification of logical variables and arbitrary number of body atoms introduce non-differentiable structures since the forward-chaining's output is non-differentiable with respect to the determination of logical variables and body atoms. Considering that the structures of existing neuro-symbolic ILP model are fully differentiable, **relaxing Horn rules into stochastic structures is one of our innovation**.
> - Simply adapting existing techniques (e.g., REINFORCE) to the ILP setting is not sufficient, we propose a set of new techniques with theoretical foundations, including
>     1. **Linear sampling of body atoms** (Algorithm 2): To sampling body atoms from FCNN’s stochastic structures, we propose a linear sampling of body atoms, which mathematically is a sampling of multiple independent and non-identical distributed Bernoulli variables under an inequality constraint.
>     The correctness is theoretically proven (Appendix A.2).
>     Although a naive rejection sampling can realize the sampling, but the time usage may be too long if the constraint probabilty is too small (i.e., very small acceptance rate).
>     2. **Unbiased gradient estimator** (Equation 7): To estimate the gradient of FCNN, we propose a gradient estimator according to FCNN’s specific stochastic structures. We theoretically prove the estimator’s unbiasedness (Proposition 4.1) and experimentally verified its effectiveness (Table 4).
>     3. **An algorithm to extract body atoms from FCNN** (Algorithm 3): We propose an algorithm for extracting rules, expecially body atoms, from FCNN, which we theoretically prove its soundness (Appendix C.3).
>     4. **A proof for FCNN's completeness** (Theorem 4.3): We prove that FCNN is complete for modeling Horn rules.
>
>
> We add a new paragraph before the final paragraph of *Introduction* to further elaborate on our technical innovations.

---

> ### Author Response · Authors · 2025-11-25
>
> > Weakness 2: Overstated contributions. The framework remains template-based, albeit with a more relaxed form than prior work. Consequently, its ability to address the open-world assumption (OWA) is still limited. For example, it is unclear whether the proposed method can discover new predicates or formulate new symbolic concepts, rather than relying on predefined templates.
>
> - Our framework is **actually not template-based**, though we did have used the word "template" to compare FCNN with existing models. Our proposed universal meta-rules is actually just an implication, the basic form of Horn rules, which enables learning of any Horn rule.
>     - Unlike the templates in previous works, which either require different templates for each task or are limited to restricted language biases, **our universal meta-rules can be instantiated to any Horn rule**, enabling full modeling ability.
>     - **The reason why we use the word "template" in the initial manuscript is that we believe this will make it easier for ILP researchers to understand FCNN and to compare FCNN with existing works.**
>     - In the revised version, we avoid using the word "template" in FCNN, since the universal meta-rules are actually not specific templates. We revise the third to last paragraph of Introduction and the first paragraph of Section 4 to avoid such confusing.
> - **The ability of discovering new predicates does not depend on whether using predefined templates**.
>     - For example, although LRI and HRI can invent new predicates, LRI requires task-specific templates and HRI uses templates with limited expressiveness.
>     - Almost the same with existing works, the predicate invention in FCNN is accomplished via predefining the number of auxiliary predicates and assigning them learnable embeddings. Relevant information can be found in the second paragraph of Section 4.1 and the second paragraph of Section 4.2.1.
> - In ILP, OWA means the background knowledge is incomplete, i.e., some true facts related to background predicates are not given in background knowledge. By this definition, **supporting OWA or not is also not related to using templates or not**. In addition, we have discussed how to enable FCNN's CWA/OWA mode in the last paragraph of Section 4.2.2.
>
>
>
> > Weakness 3: Limited impact. The experimental evaluation is restricted to ILP benchmark datasets, which are relatively small and arguably toy problems. Given that ILP represents a niche research area, the scope of the current paper appears narrow, as reflected in the limited breadth of related work. It remains uncertain—though IMHO unlikely—whether the proposed method can generalize to real-world applications.
>
> - We already discussed many real-world applications in the first paragraph of Introduction. Those applications involve various realistic domains. More applications can be found in Cropper & Dumancic (2022), Zhang et al. (2024), etc.
>     - Recently, several works even try to combine LLM (e.g., [1]) and Multimodal Large Language Models (e.g., [2]) with logical models and ILP.
> - Regarding related works:
>     - According to other reviewers' suggestions, we add some relevant works in Related Work part.
>     - We did not discuss symbolic ILP in related work (a reason why there are just a few related works), but have discussed them with neuro-symbolic ILP in the second paragraph of Introduction. Since symbolic ILPs are significantly different from neuro-symbolic ILPs, we do not include a full list of relevant works.
>     - As discussed in Introduction, neuro-symbolic ILP is an emerging area (with potential ability towards large-scale ILP). This is another reason why there are just a few related works.
> - **As an interpretable and verifiable approach, ILP deserves a line of research**, though currently its performance cannot match SOTA models in much broader and larger applications.
>
> Reference:
>
> [1] Jundong Xu, Hao Fei, Meng Luo, Qian Liu, Liangming Pan, William Yang Wang, Preslav Nakov, Mong-Li Lee, and Wynne Hsu. 2025. Aristotle: Mastering Logical Reasoning with A Logic-Complete Decompose-Search-Resolve Framework. In Proceedings of the 63rd Annual Meeting of the Association for Computational Linguistics (Volume 1: Long Papers), pages 3052–3075, Vienna, Austria. Association for Computational Linguistics.
>
> [2] Peng, Y., Liu, Y., Xia, E., Jin, Y., Dai, W. Z., Ren, Z., ... & Zhou, K. (2025). Abductive Logical Rule Induction by Bridging Inductive Logic Programming and Multimodal Large Language Models. arXiv preprint arXiv:2509.21874.

---

> ### Author Response · Authors · 2025-11-25
>
> > Question 1: Line 059, without strong language bias. What does strong mean here?
>
> Existing works, such as LRI and HRI, leverage some form of rule templates.
> The templates specify some specific language bias, such as arity $\leq 2$ and no more than two body atoms.
> Since such language bias is very limited, we call it *strong* (similar to the term *strong inductive bias*).
>
> As mentioned above, we have rephrased that paragraph in revised manuscript. This expression is now replaced with clearer expressions.
>
>
>
> > Question 2: Line 205-206, I do not really understand “ i.e., unifying the head atom according to a probability”, “Such a unification corresponds to the property that the Horn rule only allows one head atom.”
>
> 1. “ i.e., unifying the head atom according to a probability”
>     - In contrast to symbolic unification, where HEAD is only allowed to be unified with one head-candidate atom, probabilistic unification allows every head-candidate atoms to be unified with HEAD with a probabilty assigned. The probability can be interpreted as a matching score, as in HRI's predicate unification. We interpret it as a probability, since we train the FCNN via sampling-based gradient estimator.
>     - To clarify, we change this expression to "i.e., probabilistically unifying the head atoms"
> 2. “Such a unification corresponds to the property that the Horn rule only allows one head atom.”
>     - As a syntactic requirement, a Horn rule exactly has one head atom.
>     - In contrast, a Horn rule may have zero, one, or multiple body atoms.
> 3. The probability for head atom unification is a categorical distribution, i.e., only one head atom can be sampled at once. In contrast, the body atom unification allows multiple body atoms to be sampled simultaneously.
>
>
>
> > Question 3: Line 295 “After optimizing, the distributions are supposed to collapse to deterministic distributions, […]” why?
>
> We apologize for this confusing expression.
> The unification distributions do not necessarily converge to deterministic distributions, but we expect them to converge to or approximate deterministic distributions.
> We change this expression to "After optimizing, the distributions are expected to collapse to or approximate deterministic distributions".
>
> As a similar case, the predicate unification score in HRI is also expected to converge to a one-hot vector.
>
>
>
>
>
>
>
>
> Thank you again for the thorough review. We would be glad to provide any additional information if needed.

---

### Official Review · Reviewer_Pcpa · 2025-10-31

**Soundness:** 2
**Presentation:** 1
**Contribution:** 3
**Rating:** 2
**Confidence:** 3

**Summary:**

This paper proposes FCNN, a stochastic neural network that can learn logic rule. They introduce a universal meta-rule that serves as a general template for Horn rules, removing the strong language bias and manual variable assignment present in prior neural ILP systems. FCNN performs probabilistic forward-chaining reasoning and optimizes the expected reward via REINFORCE. The paper evaluates the method on both classical tasks and large-scale ILP tasks with open-world setting, where previous neural ILP systems fail to scale, and shows that FCNN outperforms prior neural ILP systems and LLMs.

**Strengths:**

1. This paper addresses the limited expressivity and strong language bias of prior neural ILP systems.
2. This method connects inductive logic learning with stochastic gradient optimization via probabilistic rule sampling and the REINFORCE estimator.
3. The paper demonstrates scalability and interpretability on both small and large-scale ILP tasks, outperforming prior neural ILP systems and LLMs.

**Weaknesses:**

1. The overall algorithm is not clearly presented, algorithm 1 and 2 describe partial components (rule sampling and body-atom extraction), but the complete training loop is missing, which makes the method section difficult to follow.
2. The theoretical part on probabilistic relaxation equivalence is intuitive but lacks formal assumptions (e.g., boundedness, convergence) and a detailed proof.
3. The efficiency and scalability of the method are not discussed, while a simple complexity estimate is given for body-atom sampling, this method could be more computationally expensive than traditional neural ILP. Its efficiency should therefore be analyzed or at least empirically compared with other neural ILP systems and LLMs.

**Questions:**

1. It is impressive that FCNN solves all tasks and outperforms other neural ILP systems, but the discussion is limited. Could the authors clarify which components (e.g., probabilistic rule sampling, variable unification, or optimization scheme) contribute most to the improvements?

2. What is its sample and computational efficiency compared to other neural ILP or LLM-based reasoning systems? How does runtime scale with the number of predicates or body-atom length?

---

> ### Author Response · Authors · 2025-11-25
>
> We sincerely thank you for the insightful comments and constructive suggestions, which are highly valuable in improving our manuscript.
> We also appreciate your recognition of the rich expressiveness, scalability, and interpretability of our model.
>
> In the following, we address your concerns point by point and detail how we revised the paper based on your suggestions.
> **(The reference number or cited texts correspond to the revised paper, which we have already uploaded to the review system.)**
>
>
>
> > Weakness 1: The overall algorithm is not clearly presented, algorithm 1 and 2 describe partial components (rule sampling and body-atom extraction), but the complete training loop is missing, which makes the method section difficult to follow.
>
> We add the pseudo-code of the complete training loop in Algorithm 1.
> Since we do not find a readable way to show pseudo-code on OpenReview, we invite you to review the pseudo-code in our revised paper.
>
>
>
> > Weakness 2: The theoretical part on probabilistic relaxation equivalence is intuitive but lacks formal assumptions (e.g., boundedness, convergence) and a detailed proof.
>
> According to your suggestions, we improve our proof for Theorem 4.3 in Appendix A.4.
> The proof now is very detailed with formal settings.
> We invite you to review the new proof in our revised paper.
>
> It is worth mentioning that the theorem is about model expressivity, but not convergence.
> To the best of our knowledge, there is no convergence analysis for existing neuro-symbolic ILP models, mainly because the models are highly non-linear.
> To the best of our knowledge, the most advanced theoretical result come from HRI, where HRI's model expressivity has been proved.
> However, HRI's model can only express very limited Horn rules, such as arity $\leq$ 2 and no more than two body atoms.
> In contrast, our model can fully express all Horn rules.

---

> > ### Author Response · Authors · 2025-11-25
> >
> > > Weakness 3: The efficiency and scalability of the method are not discussed, while a simple complexity estimate is given for body-atom sampling, this method could be more computationally expensive than traditional neural ILP. Its efficiency should therefore be analyzed or at least empirically compared with other neural ILP systems and LLMs.
> > >
> > > Question 2: What is its sample and computational efficiency compared to other neural ILP or LLM-based reasoning systems? How does runtime scale with the number of predicates or body-atom length?
> >
> > - We experimentially compare the time usage of HRI, DFORL, FCNN on 1 CPU \& 1 GPU (32GB). The selected tasks have varying difficulties in terms of program size, i.e., the total number of atoms in the solution provided in Evans & Grefenstette (2018). We invite you to review Appendix C.4 for more details.
> >     - The results reveal that DFORL is the slowest, probably because DFORL requires an expensive preprocessing for constant substitution.
> >     - FCNN and HRI take about the same amount of time on average, justifying that, although FCNN supports richer language bias, FCNN's time consumption is comparable to existing works.
> >     - We also add LLM’s time usage, but it is not comparable to others since we call the public API, and the resource behind the API is unknown (probably fairly more expensive than the resource we use).
> > - Also in Appendix C.4, we study the runtime curves with respect to different number of body atoms and number of predicates. Experimental results show that FCNN's runtime does not increase significantly due to these two factors. We invite you to see Appendix C.4 for further details.
> > - Theoretical analysis of time complexity is intractable because the most computationally expensive part is forward chaining. Borrowing the idea from HRI, the time complexity of forward chaining increases exponentially with the number of unique non-existentially quantified logical variables in rule's body. However, since the logical variables are sampled in each training step of FCNN, it is not tractable to estimate the time complexity.
> >     - Regarding the effect of the number of predicates: According to Sec. 4.1, the number of predicates linearly increases the size of the head-candidate atom $\mathcal{A}^{\rm h}$ set and the body-candidate atom set $\mathcal{A}^{\rm b}$. According to Sec. 4.2.2 \& 4.2.3, the time complexity of computing unification probability increases linearly with the size of $\mathcal{A}^{\rm h}$ or $\mathcal{A}^{\rm b}$. Consequently, the time complexity increases linearly with the number of predicates (but can be fully parallelized on GPU).
> >     - Regarding the effect of number of body atoms: Likewise, the body atoms' predicates and variables affect computation time by affecting the computation in forward chaining. Again, the logical variables and atoms are sampled in each training step of FCNN, so it is not tractable to estimate the time complexity.
> >
> > It is worth mentioning that, although we also believe analyzing time/space complexities are important, the main goal of our work is scaling up in terms of supporting richer language bias.
> > Better scaling up in terms of time/space is an interesting topic for future works.

---

> ### Author Response · Authors · 2025-11-25
>
> > Question 1: It is impressive that FCNN solves all tasks and outperforms other neural ILP systems, but the discussion is limited. Could the authors clarify which components (e.g., probabilistic rule sampling, variable unification, or optimization scheme) contribute most to the improvements?
>
> **The impressive performance of FCNN come from its richer expressiveness.**
> Let's take the task Fizz as example, which has the lowest average success rate across all baselines.
> Fizz learns the concept of "multiples of three".
> The most concise solution is $target(X) \leftarrow zero(X), target(W) \leftarrow target(X) \wedge succ(X, Y) \wedge succ(Y, Z) \wedge succ(Z, W)$.
> To infer that  "3" is a multiple of three, our solution only require two forward chaining steps.
> In contrast, the solution provided by Evans & Grefenstette (2018) requires 3 steps.
> Besides, the program size (defined as the number of atoms) of our solution is 7, and the program size of the solution in Evans & Grefenstette (2018) is 11.
> **In summary, supporting richer expressiveness may help FCNN learn more concise solution that requires fewer inference steps.**
>
> Regarding the questions which components contribute most, we cannot actually give an answer since **all the components in FCNN form an indivisible whole**.
> For example, the probabilistic unification of head atoms, body atoms, and variables take charge of sampling head atoms, body atomds, and variables, respectively, without any of it cannot sample a well-formed Horn rule.
> The optimization scheme is designed according to the stochastic nature of FCNN's structure.
> In addition, since all the components are designed for FCNN, we cannot find its counterpart in existing models.
>
>
>
>
>
> Thank you again for the thorough review. We would be glad to provide any additional information if needed.

---

### Official Review · Reviewer_WSE3 · 2025-11-10

**Soundness:** 3
**Presentation:** 3
**Contribution:** 3
**Rating:** 6
**Confidence:** 3

**Summary:**

This paper introduces FCNN, which addresses a fundamental limitation in neuro-symbolic ILP, namely that most existing methods are restricted to unary/binary predicates and ≤2 body atoms. The paper proposes universal meta-rules that probabilistically unify with candidate atoms and variables through learned embeddings, optimized via nested REINFORCE. Key contributions: (1) direct learning of variable-argument unification vs. manual specification, (2) linear-time constrained sampling algorithm, (3) support for arbitrary predicate arities. Results: 100% success on standard small benchmarks including the historically difficult Fizz/Buzz tasks; outperforms LLM on GeoILP with high-arity predicates (up to 8 arguments). Theorem 4.3 proves completeness.

**Strengths:**

Originality: Universal meta-rule framework removes restrictive templates; direct variable-argument learning vs. manual specification is novel approach. Symbol randomization for fair LLM comparison is methodologically sophisticated.

Quality: Completeness theorem provides theoretical foundation; ablation studies examine key design choices; 100% success on standard benchmarks; handles 8-arity predicates where baselines fail.

Clarity: Motivation clear; problem formulation well-justified; experimental setup rigorous.

Significance: Improves on fundamental scalability barrier in neuro-symbolic ILP; enables learning of complex rules with arbitrary arities; GeoILP benchmark specifically designed to test claimed contributions.

**Weaknesses:**

1. Figure 1 (Sensitivity of sample size) shows larger sample sizes increase variance with no explanation provided. This raises questions about RLOO baseline effectiveness and practical deployability.

2. No time/space complexity analysis, runtime comparisons, or convergence speed discussion. Critical for assessing practical scalability beyond synthetic benchmarks.

3. Symbolic hyperparameters (|U|, B, V, auxiliary predicates) replace template design bias with hyperparameter bias. No guidance for setting these; ablations only cover Na, Nv. "Sufficiently large" appears frequently without bounds.

4. CWA/OWA comparison on only 4 tasks; missing ablations for embedding dimensions, number of meta-rules, entropy decay schedule.

5. Extension from equality (Ahmed et al.) to inequality constraint claimed "almost the same" but is non-trivial technical step deserving explicit derivation.

6. GeoILP limited to "basic" level despite scalability claims; no failure mode analysis; learned rules not shown/analyzed for interpretability.

**Questions:**

How were symbolic hyperparameters chosen for GeoILP? Was domain knowledge used? What happens with significantly oversized B, V—does optimization fail or does entropy regularization compensate?

Briefly elaborate the key technical step extending equality constraint (s=k) to inequality (s≤B) beyond "almost the same" citation.

What are time/memory complexities? How do training times compare to HRI/DFORL on matched tasks?

Can you show examples from GeoILP? Are they interpretable and geometrically meaningful?

Did you test LLM with original (non-randomized) GeoILP predicates to quantify the "reasoning gap" closed by semantic knowledge?

---

> ### Author Response · Authors · 2025-11-25
>
> We sincerely thank you for the insightful comments and constructive suggestions, which are highly valuable in improving our manuscript.
> We also appreciate your recognition of the clear motivation, originality, theoretical foundation, rigorous experiments of our method.
>
> In the following, we address your concerns point by point and detail how we revised the paper based on your suggestions.
> **(The reference number or cited texts correspond to the revised paper, which we have already uploaded to the review system.)**
>
>
>
> > Weakness 1: Figure 1 (Sensitivity of sample size) shows larger sample sizes increase variance with no explanation provided. This raises questions about RLOO baseline effectiveness and practical deployability.
>
> **The high variance of larger sample sizes is due to the fewer training steps.**
> - As mentioned in the initial manuscript, to conduct fair comparison, we keep the total number of training samples the same, i.e., increasing the sample size will decrease training steps accordingly.
> - We add an experiments to justify this new claim (the fourth paragraph of Sec. 5.3 and Figure 1). In Figure 1.b \& Figure 1.d, we keep the number of training steps the same, i.e., increasing the sample size will NOT decrease training steps. In this case, a large sample size can improve the balanced accuracy and the variances typically do not increase obviously, even decrease on some tasks. This indicates that **the superior performance of large sample sizes requires sufficient training steps**.
>
> In addition, in the proposal of the vanilla RLOO (Kool et al., 2019), only 4 samples suffices.
> Figure 1 shows that 4 samples do not cause high variance in any circumstance.
>
> In practice, we recommend considering the trade-off between performance and computing resources (since larger sample sizes requrie more computing resources).
>
>
>
> > Weakness 2: No time/space complexity analysis, runtime comparisons, or convergence speed discussion. Critical for assessing practical scalability beyond synthetic benchmarks.
> >
> > Question 3: What are time/memory complexities? How do training times compare to HRI/DFORL on matched tasks?
>
> - We experimentially compare the time usage of HRI, DFORL, FCNN on 1 CPU \& 1 GPU. The selected tasks have varying difficulties in terms of program size, i.e., the total number of atoms in the solution provided in Evans & Grefenstette (2018). We invite you to review Appendix C.4 for more details.
>     - The results reveal that DFORL is the slowest, probably because DFORL requires an expensive preprocessing for constant substitution.
>     - FCNN and HRI take about the same amount of time on average, justifying that, although FCNN supports richer language bias, FCNN's time consumption is comparable to existing works.
> - Theoretical analysis of time complexity is intractable because the most computationally expensive part is forward chaining. Borrowing the idea from HRI, the time complexity of forward chaining increases exponentially with the number of unique non-existentially quantified logical variables in rule's body. However, since the logical variables are sampled in each training step of FCNN, it is not tractable to estimate the time complexity.
>
> It is worth mentioning that, although we also believe analyzing time/space complexities are important, the main goal of our work is scaling up in terms of supporting richer language bias.
> Better scaling up in terms of time/space is an interesting topic for future works.
>
>
>
> > Weakness 3: Symbolic hyperparameters (|U|, B, V, auxiliary predicates) replace template design bias with hyperparameter bias. No guidance for setting these; ablations only cover Na, Nv. "Sufficiently large" appears frequently without bounds.
> >
> > Weakness 4: CWA/OWA comparison on only 4 tasks; missing ablations for embedding dimensions, number of meta-rules, entropy decay schedule.
>
> - According to your suggestions, we add comprehensive hyperparameter analysis in Appendix C.3 for all the hyperparameters you mentioned. We invite you to see the revised paper for details.
> - We also add experiments of CWA/OWA comparison on all tasks in Appendix C.2. The main conclusion remains the same, i.e., the CWA mode is slightly better since the OWA mode suffers from a larger hypothesis space.
> - In Appendix A.4, "Sufficiently large" has been replaced with formal conditions.

---

> > ### Author Response · Authors · 2025-11-25
> >
> > > Weakness 5: Extension from equality (Ahmed et al.) to inequality constraint claimed "almost the same" but is non-trivial technical step deserving explicit derivation.
> > >
> > > Question 2: Briefly elaborate the key technical step extending equality constraint (s=k) to inequality (s≤B) beyond "almost the same" citation.
> >
> > We add a complete proof in Appendix A.2 and invite you to review the revised proof in the revised paper.
> >
> >
> >
> > > Weakness 6: GeoILP limited to "basic" level despite scalability claims; no failure mode analysis; learned rules not shown/analyzed for interpretability.
> >
> > > Question 4: Can you show examples from GeoILP? Are they interpretable and geometrically meaningful?
> >
> > - Although we only tested on *basic* level, it is already large enough towards existing methods (e.g., predicate arity, number of body atoms) and is unreachable by existing methods (Chen et al., 2025).
> > **Therefore, our main contribution is well-justified by FCNN's superior performance on GeoILP's *basic* level.**
> > - It is not really tractable to analyze failure mode since FCNN cannot perfectly solve nearly any GeoILP task (although FCNN achieves the SOTA balanced accuracy).
> > - The learned rules are intepretable since they are Horn rules. However, we found the generated rules not very geometrically mearningful. We argue this is because GeoILP's basic level is generated by relatively trivial rules.
> >     - For example, on perp\_1, FCNN found the following rule: $perp(X0, X1, X1, X1) \leftarrow coll(X0, X2, X3)$. This implication is not wrong but not very helpful.
> >
> >
> >
> > > Question 1: How were symbolic hyperparameters chosen for GeoILP? Was domain knowledge used? What happens with significantly oversized B, V—does optimization fail or does entropy regularization compensate?
> >
> > All the symbolic hyperparameters for GeoILP are chosen manually. We almost never tuned them carefully and all the GeoILP tasks share the same hyperparameters.
> > The only priors for setting the hyperparameters are from small-scale benchmarks and we slightly *scale up* the hyperparameters (e.g., slightly increasing the number of unversal meta-rules).
> >
> > We believe that more caraful hyperparameter tuning can further improve the performance. We did not conduct such tuning since FCNN already achieves SOTA performance.
> >
> > In addition, as mentioned before, we have added hyperparameter analysis in Appendix C.3. We invite you to see that section for further analysis on $B$ and $V$.
> >
> >
> >
> > > Question 5: Did you test LLM with original (non-randomized) GeoILP predicates to quantify the "reasoning gap" closed by semantic knowledge?
> >
> >
> >
> > We believe this is a good suggestion for developping LLM-based ILP learner.
> > However, our work focus on developping neuro-symbolic ILP learner, and we did not conduct such experiment.
> >
> >
> >
> >
> >
> >
> > Thank you again for the thorough review. We would be glad to provide any additional information if needed.

---

### Author Response · Authors · 2025-11-25
**Summary of revision**

Dear Reviewers and Area Chairs,

We sincerely thank you for reviewing our submission. This message is about the main revisions in our submission.

- Improved theoretical proofs: We have revised the proofs in Appendix A.2 \& A.4 to make them complete and formally derived.
- More experiments:
    - In Section 5.3, we add experiments to explain the high variance in Figure 1.
    - In Appendix C.3, we add a comprehensive analysis of hyperparameters.
    - In Appendix C.4, we add runtime analysis.
- Clearer presentations:
    - In Algorithm 1, we add a pseudo-code to illustrate the entire training procedure.
    - In the last three paragraphs of Introduction and in the first paragraphs of Section 4, we further clarify our innovation and add a top-down description of our proposed model.
    - We add more related works in Section 2.
    - We revise all the typos we found.

Best regards,

All authors

---

### Meta-Review · Area_Chair_TKPe · 2026-01-05

**Summary:**

Reviewers agreed that the paper targets an important problem in neuro-symbolic ILP, but raised concerns about limited novelty, as the approach largely extends existing probabilistic relaxation and REINFORCE-based methods. The claimed ability to address open-world reasoning and predicate invention was viewed as overstated, given the continued reliance on predefined structures and hyperparameters. In addition, issues with clarity, scalability, and limited ablation analysis reduce confidence in the strength and generality of the contributions. Overall, this paper does not meet the acceptance bar of ICLR.

**Reviewer Concerns:**

Although the authors provided a detailed rebuttal, the responses regarding contribution, presentation, and related concerns were not sufficient to flip the reviewers’ leaning toward rejection.

**Reviewer Scores:**

The reviewers whose ratings are below 6 may raise the rating, but hardly to show a leaning toward acceptance.

---

### Decision · Program_Chairs · 2026-01-26

Reject